

# Customized Deep Learning for Precipitation Bias Correction and Downscaling

Fang Wang[1], Di Tian[1]*, and Mark Carroll[2]

[1]Department of Crop, Soil, and Environmental Sciences, Auburn University, Auburn, AL 36849, USA
[2]Computational and Information Science Technology Office, NASA Goddard Space Flight Center Greenbelt, MD 20771, USA

*Correspondence to: Di Tian (tiandi@auburn.edu)

**Abstract.** Systematic biases and coarse resolutions are major limitations of current precipitation datasets. Many deep learning (DL) based studies have been conducted for precipitation bias correction and downscaling. However, it is still challenging for the current approaches to handle complex features of hourly precipitation, resulting in incapability of reproducing small scale features, such as extreme events. This study developed a customized DL model by incorporating customized loss functions, multitask learning, and physically relevant covariates to bias correct and downscale hourly precipitation data. We designed six scenarios to systematically evaluate the added values of weighted loss functions, multi-task learning, and atmospheric covariates compared to the regular DL and statistical approaches. The model was trained and tested using the Modern-Era Retrospective analysis for Research and Applications version 2 (MERRA2) reanalysis and the Stage IV radar observations over northern coastal region of Gulf of Mexico. We found that all the scenarios with weighted loss functions performed notably better than the other scenarios with conventional loss functions and a quantile mapping-based approach at hourly, daily, and monthly time scales as well as extremes. Multitask learning showed improved performance on capturing hourly precipitation climatology, aggregated precipitation at daily and monthly scales, and detailed features of extreme events, while the improvement is not as large as from weighted loss functions. Accounting for atmospheric covariates further improved the model performance for capturing extreme events. We show that the customized DL model can better downscale and bias correct precipitation datasets and provide improved precipitation estimates at fine spatial and temporal resolutions where regular DL and statistical methods experiencing challenges.

## 1 Introduction

Precipitation is a major component of hydrological cycle and is fundamentally important for many applications such as water resources planning and management, disaster risk management, agriculture, amongst many others. Due to limited coverage of ground-based rain gauges, numerous gridded precipitation datasets have been developed over the past decades, including gauge-based, satellite-based, reanalysis products, and merged products (Beck, Wood, et al., 2019; Sun et al., 2018). These datasets are different in terms of data sources, coverage, spatial and temporal resolution and algorithms (see Sun et al., 2018





for a review), which provide a potential source of information to regions where conventional in situ precipitation
measurements are lacking (Sun et al., 2018).

Gridded precipitation data have proven to be useful across a wide range of research fields, including climate trend and
extreme precipitation (Bhattacharyya et al., 2022; DeGaetano et al., 2020; Fischer & Knutti, 2016; I.-W. Kim et al., 2019;
King et al., 2013),  droughts and floods monitoring (Aadhar & Mishra, 2017; Peng et al., 2020; Suliman et al., 2020; Zhong
et al., 2019), and driving hydrological models (Raimonet et al., 2017; H. Xu et al., 2016). However, many studies have
identified that these gridded precipitation datasets include substantial biases in certain aspects compared to in situ
observations (Aadhar & Mishra, 2017; Ashouri et al., 2016; Bitew & Gebremichael, 2011; Cavalcante et al., 2020; Jiang et
al., 2021; Jury, 2009; Rivoire et al., 2021; Sun et al., 2018; K. Tong et al., 2014; H. Xu et al., 2016; Yilmaz et al., 2005). For
example, Ashouri et al. (2016) evaluated the performance of NASA's Modern-Era Retrospective Analysis for Research and
Applications (MERRA) precipitation reanalysis dataset and found that MERRA tends to overestimate the frequency at which
the 99th percentile of precipitation is exceeded and underestimate the magnitude of extremes, especially over the Gulf Coast
regions of the United States. Furthermore, spatial resolution for most of these gridded precipitation datasets is relatively
coarse for local scale applications (mostly above 0.25°, see Sun et al., 2018). Therefore, the gridded precipitation datasets
require bias correction and downscaling (Duethmann et al., 2013; Emmanouil et al., 2021; Mamalakis et al., 2017; Seyyedi
et al., 2014).

Bias correcting and downscaling gridded precipitation data is challenging due to its complex characteristics (e.g., highly
skewed, unbalanced feature, and complex spatial-temporal structure). Various approaches have been developed to tackle this
issue including traditional quantile mapping (QM) based bias correction and downscaling methods (e.g., Cannon et al., 2015;
H. A. Panofsky & G. W. Brier, 1968; Thrasher et al., 2012; Wood et al., 2002) and recent machine learning based
approaches such as random forests (X. He et al., 2016; Legasa et al., 2022; Long et al., 2019; Mei et al., 2020; Pour et al.,
2016), support vector machines (Tripathi et al., 2006) and artificial neural networks (Schoof & Pryor, 2001; Vandal et al.,
2019). Recently, advances in deep learning have made a significant impact on many fields and have been proven superior to
traditional machine learning methods because of their powerful abilities in learning spatiotemporal feature representation in
an end-to-end manner (Ham et al., 2019; Reichstein et al., 2019; Shen, 2018). In particular, deep learning (DL) with
convolutional neural network (CNN) types of approaches have achieved notable progress in modeling spatial context data
(LeCun et al., 2015) and have been used for bias correcting and downscaling low spatial resolution data (Kumar et al., 2021;
Sha et al., 2020a, 2020b; Vandal et al., 2018; Wang et al., 2021; M. Xu et al., 2020), climate model outputs (François et al.,
2021; Liu et al., 2020; Pan et al., 2021; Rodrigues et al., 2018; Wang & Tian, 2022), reanalysis products (Baño-Medina et
al., 2020), and weather forecast model outputs (Harris et al., 2022; W. Li et al., 2022). While these studies have indicated
many promising strengths and advantages over traditional downscaling and bias correction approaches, most of them have
difficulties to capture local-, small-scale features such as extremes for unseen dataset. For example, Baño-Medina et al.





(2020) designed different DL configurations with different number of plain CNN layers to bias correct and downscale daily ERA5-Interim reanalysis from 2° spatial resolution to 0.5°, and the overall performance is still marginal compared with simple generalized linear regression models and highly underestimated precipitation extremes. Harris et al. (2022) developed a generative adversarial networks (GANs) architecture to bias correct and downscale weather forecast outputs and found that

it is more challenging to account for forecast error (or bias) in a spatially-coherent manner compared to pure downscaling problem (Kumar et al., 2021; Sha et al., 2020a, 2020b; Vandal et al., 2018; Wang et al., 2021; M. Xu et al., 2020). The reason for that may be due to the sparsity of training data on extreme events. Deep learning (DL) models, however, need large training data in order to obtain a better regularization model for rare events in the unseen dataset.

Customized DL models have been proposed to generate physically consistent results and have better generalization ability

for out of pocket dataset in the earth and environmental science field, which include incorporating customized loss functions (Kashinath et al., 2021), inputs from physically relevant auxiliary predictors (i.e., covariates) (W. Li et al., 2022; Rasp & Lerch, 2018), and customized multitask learning (Ruder, 2017). For example, Daw et al. (2017) indicated success in lake temperature modeling by incorporating a physics-based loss function in the DL objective compared to regular loss function. W. Li et al. (2022) used CNN-based approach to postprocess numerical weather prediction model output and found that the

use of auxiliary predictors greatly improved model performance compared with raw precipitation data as the only predictor. A multitask model is trained to predict multiple tasks that are driven by the same underlying physical processes, thus has the potential to learn to better represent the shared physical process and better predict the variable of interest (Ruder, 2017). Multitask models have proven effective in several applications including natural language processing (D. Chen et al., 2014; Seltzer & Droppo, 2013), computer vision (Girshick, 2015), as well as hydrology (Sadler et al., 2022). In addition, most of

the previous bias correction and downscaling studies focused on daily time scale (Baño-Medina et al., 2020; François et al., 2021; Harris et al., 2022; Kumar et al., 2021; Liu et al., 2020; Pan et al., 2021; Rodrigues et al., 2018; Sha et al., 2020b; Vandal et al., 2018; Wang et al., 2021). However, the distribution of hourly precipitation data within a day is more important than daily or monthly aggregations for impacts and risks from warming-induced precipitation changes (Yang Chen, 2020). Customized deep learning, through incorporating customized loss functions, covariates, or customized multitask learning,

have the potential to fundamentally improve hourly precipitation bias correction and downscaling.

In this study, we will explore customized DL for precipitation bias correction and downscaling, aiming to make a step forward to addressing the current challenges described above. We designed a set of experiments to address this hypothesis using the Modern-Era Retrospective analysis for Research and Applications Version 2 (MERRA2) reanalysis and the Stage IV radar precipitation data. The structure of this paper is organized as follows: Section 2 introduced data and methodology,

including the deep learning architecture and experimental designs for different scenarios, and a traditional bias correction approach as a benchmark; Section 3 presents results; discussion and conclusions are provided in Section 4 and 5, respectively.



## 2 Data and methodology

### 2.1 Data and study area

MERRA2 is a state-of-the-art global reanalysis product generated by the NASA Global Modeling and Assimilation Office (GMAO) using the Goddard Earth Observing System, version 5 (GEOS-5), and was introduced to replace and extend the original MERRA dataset (Reichle et al., 2017). It incorporates new satellite observations through data assimilation and benefits from advances in the GEOS-5 (Reichle et al., 2017). There are two hourly total precipitation ($P$) data available from MERRA2 reanalysis product, the model analyzed precipitation computed from the atmospheric general circulation model

and the observation-corrected $P$ (Reichle et al., 2017). Both have a spatial resolution of 0.5° in latitude and 0.625° in longitude (~50km). MERRA2 observation-corrected precipitation have been used extensively in hydro-climatological analysis and modelling (Yingying Chen et al., 2021; Hamal et al., 2020; X. Xu et al., 2019; X. Xu et al., 2022). However, it still suffers from substantial biases (e.g., Hamal et al., 2020; X. Xu et al., 2019). This study will bias correct and downscale MERRA2 observation-corrected $P$ using the Stage IV radar data (Y. Lin & Mitchell, 2005) from the National Centers for

Environmental Prediction (NCEP) as the observational reference. The Stage IV radar data has a 4 km spatial and hourly temporal resolution and covers the period from 2002 until the near present (2021 in this study). Stage IV radar was generated by merging data from 140 radars and about 5500 gauges over the continental United States (Y. Lin & Mitchell, 2005; Nelson et al., 2016). The Stage IV provides highly accurate $P$ estimates and has therefore been widely used as a reference for evaluating other $P$ products (e.g., A AghaKouchak et al., 2011; Amir AghaKouchak et al., 2012; Beck, Pan, et al., 2019;

Habib et al., 2009; Hong et al., 2006; Nelson et al., 2016; Zhang et al., 2018). The Stage IV dataset is a mosaic of regional analyses produced by 12 River Forecast Centers (RFCs) and is thus subject to the gauge correction and quality control performed at each individual RFC (Nelson et al., 2016).

     The bias correction and downscaling experiments were performed in the rectangle coastal area of Gulf of Mexico covering the entire states of Alabama, Mississippi and Louisiana, and parts of neighbour states in the United States, ranging

from -94.375° to -85.0° in longitude and from 29.0° to 35.0° in latitude. The study area falls into the humid subtropical climate and is highly influenced by extreme $P$ events such as convective storms and hurricanes.

### 2.2 Customized DL approaches

#### 2.2.1 Overview

     This section firstly presents a brief description of a DL approach, namely, Super Resolution Deep Residual Network

(SRDRN). Then, multitask learning and customized loss functions are introduced based on the SRDRN architecture to construct customized DL approaches. Finally, we designed different modeling experiments, which include different





combinations of multitask learning, customized loss functions, and *P* covariates as predictors, in order to evaluate the added values of each component of the customized DL approaches.

### 2.2.2 SRDRN model

The SRDRN model is an advanced deep CNN type architecture and has been tested for downscaling daily *P* and temperature through synthetic experiments (Wang et al., 2021) and for bias correcting near surface temperature simulations from global climate models (Wang & Tian, 2022), considerably outperforming the conventional approaches. Here we provide a brief description of the SRDRN algorithm. For more details, the readers may refer to Wang et al. (2021). The SRDRN algorithm was developed based on a novel super scaling deep learning approach in the computer vision field (Ledig

et al., 2017). Basically, the SRDRN algorithm is comprised of residual blocks and upsampling blocks with convolutional and batch normalization layers. For feature extraction, the convolutional layers apply filters to go through the input data to build a local connection within nearby grids by computing the element-wise dot product between the filters and different patches of the input. The outcome is followed by a nonlinear activation function, here parametric ReLU (He et al., 2015) in this study. Batch normalization is a technique to standardize the inputs to a layer for each mini-batch so that the learning process

can be stabilized and the training of the model can be accelerated (Ioffe & Szegedy, 2015).

    With convolutional and batch normalization layers, the residual blocks are designed to extract fine spatial features while avoid degradation issue for the very deep neural network. Compared to plain CNN architectures, residual blocks can improve the performance of extensively deep networks (Silver et al., 2017) without suffering from model accuracy saturation and degradation (K. He et al., 2016) because residual blocks execute residual mapping and include skipping connections. In

this study, the way that skipping connection skips layers and connects next layers is through element-wise addition. The total number of 16 residual blocks were used in the SRDRN architecture, which makes the network very deep and able to extract fine spatial features.

    The upsampling blocks are applied to increase spatial resolution for downscaling purpose. The upsampling process is executed directly on the feature maps generated from the residual blocks and each upsampling block is composed of one

convolutional layer and one upsampling layer followed by parametric ReLU activation function. The defaulted nearest neighbor interpolation was chosen in the upsampling layers to increase spatial resolution. Each upsampling block sequentially and gradually increases the input low resolution feature maps by a factor of 2 or 3. In this study, the downscaling ratio (the ratio between coarse resolution and high-resolution data) is 12 and thus we used 3 upsampling blocks with two blocks having a factor of 2 and one block having a factor of 3.

### 2.2.3 SRDRN model with multitask learning





We included an additional *P* classification task in the SRDRN model. Besides bias correcting and downscaling continuous hourly *P* values as a primary task, we added another task to bias correct hourly *P* categories. Since these two tasks are highly relevant to each other, it is expected that the classification task can improve the model performance on bias correcting and downscaling *P*.

Specifically, for the SRDRN with multitask learning, one convolutional layer (256 filters and 3x3 kernels) follows the last element-wise addition operation to summarize feature maps, then the architecture splits into two sections (Figure 1). The first section with two additional convolutional  layers (first one with 64 filters and the second with 4 filters) followed by the Softmax activation (Goodfellow et al., 2016) is used for bias correcting *P* categories as a multiclass classification task and the other section with upsampling blocks is used for the purpose of bias correcting and downscaling hourly *P*. The

classification task classifies the hourly *P* at each grid into four categories: 0-0.1mm/h as no rain, 0.1-2.5mm/h as light rain, 2.5-10mm/h as moderate rain and >10mm/h as heavy rain (Tao et al., 2016). Due to radar sensors' uncertainty in very light rainfall, 0.1 mm/h is commonly used as a threshold to determine if there is rain (Tao et al., 2016). Since the classification task is executed on the feature maps at the coarse resolution, we aggregated Stage IV *P* (namely, coarsened Stage IV in this study) into the same spatial resolution as MERRA2 and classified the upscaled *P* data into the four groups as target labels.

[Insert Figure 1]

### 2.2.4 Customized loss functions

We developed a weighted mean absolute error (MAE) loss function ($L_{MAE\_weighted}$) to balance precipitation data where weights change with precipitation values as shown below,

$$L_{MAE\_weighted} = \frac{\sum_{i=1}^{n} w_1 \cdot |y_{pred} - y_{true}|}{n} \tag{1}$$

where *n* is the total number of grids in a batch, $w_1$ is the weight for each absolute error between model predicted value $y_{pred}$ and the true value $y_{true}$. The weight $w_1$ changes with the actual true value $y_{true}$,

$$w_1 = \begin{cases} MIN & y_{true} \leq MIN \\ y_{true} & MIN < y_{true} < MAX \\ MAX & y_{true} \geq MAX \end{cases}$$

where *MIN* is the lowest threshold and  *MAX* is the highest threshold for the weights. In other words, when the $y_{true}$ value is below (above) *MIN* (*MAX*), $w_1$ equals *MIN* (*MAX*), otherwise $w_1$ equals $y_{true}$ itself. Thus, loss is weighted directly by the

*P* value at grid cell scale, which has been proven more effective than weighted by *P* bins (Ravuri et al., 2021; Shi et al., 2017). Note that all of the gridded *P* data including Stage IV and MERRA-2 are logarithmically transformed [i.e.,





$y=log(x+1)$] in order to amplify the normality and reduce the skewness of $P$ data (Sha et al., 2020b). In Equation 1, $y_{true}$ and $y_{pred}$ are transformed $P$ values. *MIN* was set to *log(0.1+1)* and *MAX* was set to *log(100+1)*, where maximum 100mm/h was chosen as the highest threshold before log transformation for robustness to spuriously large values in the Stage IV radar (Ravuri et al., 2021) and 0.1 mm/h is commonly used as a threshold to determine if there is rain for radar data (Tao et al., 2016).

For the four $P$ categories, most data fall into the no rain category (over 88% in the coarsened Stage IV) and minority data fall into the heavy rain category (about 0.2% in the coarsened Stage IV). Thus, handling class-imbalance is of great importance in this situation, where the minority class for heavy rain category is the class of most interest with respect to this learning task. The regular cross entropy loss function for classification task could result in underestimation of minority class (Fernando & Tsokos, 2021). Thus, we applied a weighted cross entropy as loss function ($L_{weighted\ Cross-entropy}$) for the classification task in order to penalize more towards heavy rain category as follows,

$$L_{weighted\ Cross-entropy} = -\sum_{i=1}^{n}\sum_{j=1}^{k} w_{2,j} \cdot p(y_{i,j}) \cdot \log(q(y_{i,j})) \tag{2}$$

where $w_{2,j}$ denotes the weight for the $j$th class, $p(y_{i,j})$ represents the true distribution of the $i$th grid for the $j$th class, and $q(y_{i,j})$ represents the predicted distribution. $k$ is the number of classes (equals to 4 in this study). $w_{2,j}$ was set to 1, 5, 15 and 80 for no rain, light rain, moderate rain, and heavy rain classes, respectively, which is roughly based on the opposite percentage (i.e., 1, 5, 15, 80 are approximately from the percentages of heavy, moderate, light and no rain categories, respectively) for each category of the coarsened Stage IV.

**2.2.5 Experiment Design**

To comprehensively evaluate the added value of each component of customized DL models including weighted loss function, multitask learning and adding covariates, we designed six scenarios (Scenario1 to Scenario6 in Table 1). Scenario1 is based on the basic SRDRN architecture with hourly $P$ from MERRA2 as coarse-resolution input, $P$ from Stage IV as high-resolution labelled data, and regular MAE as loss function, which represents regular DL. Scenario2 is the same as Scenario 1 except using weighted MAE loss function [Eqn. (1)]. The number of trainable parameters is the same for Scenario1 and Scenario2. Scenario3 includes the classification task and the total loss is the combination of Eqn. (1) and Eqn. (2) with a weight $\lambda$ [see Eqn. (3) below], where $\lambda$ was set to 0.01 to ensure the two parts of the losses are in the same magnitude. The trainable parameters for Scenario3 increases by 30% compared to Scenario1 and Scenario2.

$$L = L_{MAE\_weighted} + \lambda \cdot L_{weighted\ Cross-entropy} \tag{3}$$

[Insert Table 1]





The other three scenarios also consider atmospheric covariates of *P* from MERRA2 as predictors, which include geopotential height, specific humidity, air temperature, eastward wind, northward wind at three different vertical levels (250, 500, 850 hPa) (e.g., Baño-Medina et al., 2020; Rasp & Lerch, 2018) as well as vertical wind (e.g., Trinh et al., 2021) at 500 hPa (OMEGA500), sea level pressure and 2 meter air temperature in single level (e.g., Panda et al., 2022; Rasp & Lerch, 215 2018) (see Table 2). For each covariate listed in Table 2, data normalization was executed as a data preprocessing step. Specifically, each covariate was normalized by subtracting the mean (μ) and dividing by the standard deviation (σ). Here μ and σ are scalar values that were calculated based on the flattened variable for the training dataset. During the testing period, model prediction was made with the normalized test dataset from MERRA2 with μ and σ calculated from the statistics of the coarse-resolution data during the test period to preserve nonstationary. Scenario4 only included atmospheric 220 covariates without using coarse resolution *P* as input and used Eqn. (1) as loss function to test whether only covariates are sufficient for estimating hourly *P*. The number of trainable parameters for Scenario4 is about 1% more compared to Scenario1 and Scenario2. Scenario5 is the same as Scenario4 except including *P* as a predictor besides atmospheric covariates and the number of trainable parameters is very close to Scenario4. Scenario6 is the same as Scenario5 except including classification task with Eqn. (3) as loss function and the number of trainable parameters is similar to Scenario3 225 (31% greater than scenarios with no multitask learning).

[Insert Table 2]

The Adam optimization algorithm was applied to train the six DL scenarios with a learning rate of 0.0001 and other default values. We found that the learning rate of 0.0001 worked stably in this study through a series of experiments. The batch size for each epoch was set to 64, and the number of epochs was set to 150 for each scenario listed in Table 1. We 230 frequently saved models and evaluated their performance with a validation dataset in order to choose the best model for prediction on the test dataset. The training process was executed using NVIDIA V100 GPU provided by the NASA High-End Computing (HEC) Program through the NASA Center for Climate Simulation (NCCS) at Goddard Space Flight Center (https://www.nccs.nasa.gov/systems/ADAPT/Prism).

At the time when we conduct this study, MERRA-2 and Stage IV hourly *P* data have 20-year overlapping period from 235 2002 to 2021. We used the first 14 years (2002 to 2015) as training dataset, the middle 3 years (2016 to 2018) as validation dataset and the more recent 3 years (2019 to 2021) as test dataset. Figure 2 shows the hourly mean or climatology for MERRA-2 and Stage IV for training and test datasets, as well as the mean differences between the test and the training periods. We can tell that there are large climatology differences (or biases) between MERRA-2 and Stage IV both for training and test datasets especially around the coastal area. Wetter conditions are observed in most of the study area in the 240 test period (average 0.03 mm/h) comparing with the training period. This allows us to assess the extrapolation capabilities of the different methods, which is particularly relevant in a changing climate.



[Insert Figure 2]

## 2.3 Statistical approach

We used a widely accepted quantile delta mapping (QDM) as a benchmark approach for *P* bias correction. Compared
to the regular quantile mapping method (H. Panofsky & G. Brier, 1968; Thrasher et al., 2012; Wood et al., 2002), QDM
accounts for the difference between historical and future climate data (here, training and test periods) and thus is capable of
preserving trend of the future climate (Cannon et al., 2015), which is critical for this study since there are substantial
differences between the precipitation during the training (2002 to 2015) and test (2019 to 2021) periods (see Figure 2). This
approach has been widely used to bias correct climate variables including *P*, which indicated better performance compared
to the other bias correction approaches (Cannon et al., 2015; Eden et al., 2012; S. Kim et al., 2021; Tegegne & Melesse,
2021; Y. Tong et al., 2021). To be specific for QDM, the bias corrected value $\hat{x}_{m,p}(t)$ for modeled data in the future
projection at time $t$ is given by applying the relative change $\Delta_m(t)$ multiplicatively to the historical bias corrected value
$\hat{x}_{o:m,h:p}(t)$,

$$\hat{x}_{m,p}(t) = \hat{x}_{o:m,h:p}(t) \cdot \Delta_m(t) \tag{4}$$

where $\hat{x}_{o:m,h:p}(t) = F_{o,h}^{-1}\left[\tau_{m,p}(t)\right]$ and $\Delta_m(t) = \frac{x_{m,p}(t)}{F_{m,h}^{-1}\left[\tau_{m,p}(t)\right]}$. $x_{m,p}(t)$ represents uncorrected modeled data in the projection
period and $\tau_{m,p}(t)$ is the percentile of $x_{m,p}(t)$ in the empirical cumulative density function (*F*) formulated by the modeled
data in the projection period over a time window around $t$. $F_{o,h}^{-1}\left[\tau_{m,p}(t)\right]$ means applying inverse empirical cumulative
density function formulated by the observed data in the historical period for $\tau_{m,p}(t)$ to obtain bias corrected value [i.e.,
$\hat{x}_{o:m,h:p}(t)$]. Similarly, $F_{m,h}^{-1}\left[\tau_{m,p}(t)\right]$ denotes applying inverse empirical cumulative density function formulated by the
260 modeled data in the historical period for $\tau_{m,p}(t)$. The time window to construct the empirical cumulative density function
around time $t$ was set to be 45 days to preserve the seasonal cycle. In this study, the historical and projection periods
correspond to the training and test data periods, respectively. The modeled and observed data correspond to MERRA2 and
coarsened Stage IV data, respectively. Details about this method are referred to Cannon et al. (2015).

The QDM bias correction was performed at the spatial resolution of MERRA2. The QDM biased corrected *P* data at
265 the coarse resolution was then bilinear interpolated into the high resolution, same as the spatial resolution of Stage IV. This
process of QDM and bilinear interpolation (X. He et al., 2016) is named as QDM_BI in the following sections.

## 2.4 Evaluation approaches

We evaluated model performance in different temporal scales including hourly and aggregated (daily and monthly)
time scales. The agreements between the observed and estimated (i.e., bias corrected and downscaled) *P* for the different





scales and extremes were quantified using the Kling-Gupta efficiency (KGE). The KGE is an objective performance metric combining correlation, bias, and variability, which was introduced in Gupta et al. (2009) and modified in Kling et al. (2012). KGE has been widely used for evaluating different datasets with observations (e.g., Beck, Pan, et al., 2019; Beck, Wood, et al., 2019; Wang et al., 2021) and as the standard evaluation metric in hydrology (Beck et al., 2017; Harrigan et al., 2018; Harrigan et al., 2020; P. Lin et al., 2019). The KGE is defined as follows:

$$KGE = 1 - \sqrt{(r-1)^2 + (\beta-1)^2 + (\gamma-1)^2} \tag{5}$$

where the correlation component $r$ is represented by correlation coefficient, the bias component $\beta$ represented by the ratio of estimated and observed means, and the variability component $\gamma$ represented by the estimated and observed coefficients of variation:

$$\beta = \frac{\mu_s}{\mu_o} \quad \text{and} \quad \gamma = \frac{\sigma_s/\mu_s}{\sigma_o/\mu_o} \tag{6}$$

where $\mu_s$ and $\mu_o$ denote the distribution mean for the estimates and observations, and $\sigma_s$ and $\sigma_o$ denote the standard deviation for the estimates and observations, respectively. KGE, $r$, $\beta$ and $\gamma$ represent perfect agreement when they equal one. In addition to KGE, the root mean square error (RMSE) and mean absolute error (MAE) metrics are also reported since they were often used to evaluate model performance on bias correction and downscaling (e.g., Maraun et al., 2015; Rodrigues et al., 2018).

To understand the performance on capturing $P$ extremes, we assessed hourly $P$ at 99[th] percentile and annual maximum wet spell in hours as well as an extreme hurricane event occurred during the test period. These extreme indices and events are highly relevant to flooding (Pierce et al., 2014) and have great environmental impact as well as impacts on property and human life.

Moreover, we evaluated $P$ classification results from Scenario3 and Scenario6, the scenarios with multitask learning for bias correcting $P$ categories, by comparing with the four categories from the coarsened Stage IV observations. The four categories from the coarsened Stage IV were generated manually based on the ranges of the four classes. We also classified the results from QDM and raw MERRA2 into the four categories and compared the results with the categories from the coarsened Stage IV. A widely used metric, namely, Intersection over Union (IoU) (Z. Li et al., 2021), is applied to evaluate classification performance, which is defined by:

$$IoU = \frac{TP}{TP+FP+FN} \cdot 100 \tag{7}$$





where *TP* represents true positives, *FP* represents false positives and *FN* represents false negatives. IoU ranges from 0 to 100 and specifies the percentage of the amount of overlap between predicted and ground truth bounding box.

## 3 Results

In this section, we present the performance of the six DL scenarios and the benchmark approach QDM_BI on bias correcting and downscaling hourly *P*, evaluated against Stage IV precipitation data during the test period from 2019 to 2021.

### 3.2 Overall agreement

The overall agreement between the observed and estimated *P* was quantified with KGE [Eq. (5)] as well as each component of KGE, which were calculated on an hourly basis for the entire test period (2019 to 2021) and for all the grid cells over the study region. Table 3 shows that Scenario2 to Scenario6 have much higher KGE than Scenario1, indicating that weighted loss function improved model performance through rebalancing hourly *P* data. Scenario1, however, highly overestimated the variability (i.e., $\gamma$ is much greater than 1) and underestimated the mean (i.e., $\beta$ is much smaller than 1), resulting in a negative KGE value. This indicates that using a regular loss function (i.e., MAE) tends to underestimate hourly *P*. The KGE values are comparable for all the scenarios using the weighted loss function. The best KGE is obtained by Scenario5, with Scenario4 and Scenario6 performing consistently well in terms of KGE, which indicates that including atmospheric covariates as predictors further improved the model performance. However, the DL and benchmark approaches performed considerably worse in terms of the correlation component $r$ of KGE than the other components (i.e., $\beta$ and $\gamma$), suggesting that long-term climatological *P* statistics are relatively easier to capture than hour to hour *P* dynamics (i.e., $r$). The benchmark, QDM_BI, also highly overestimated the variability, and has a lower KGE score than Scenario4, Scenario5 and Scenario6 of the DL approaches.

[Insert Table 3]

Table 3 also reports the results of RMSE and MAE, which are widely used to evaluate model performance on bias correction and downscaling. However, these two metrics are inadequate for pixel-wise comparison particularly when comparing two datasets with spatial biases, due to the well-known "double penalty problem" (Harris et al., 2022; Rossa et al., 2008). Specifically, for using RMSE or MAE metrics, model estimates that correctly capture the right amounts of rain in slightly incorrect locations often score worse than estimates of no rain at all. For example, Scenario1 has the lowest RMSE and MAE, but it highly underestimated the average (i.e., β is much lower than 1), while it is the worst one in all the scenarios including QDM_BI in terms of KGE scores. This illustrates the limitations of the grid point-based error like RMSE and MAE as evaluation metrics.

### 3.3 Hourly Climatology





Due to climate change, the climatology of hourly *P* over the test period (2019 to 2021) is much higher than the training period (2002 to 2015) (Figure 2). We evaluated the long-term mean (i.e., climatology) during the test period (Figure 3 and Figure 4a), which allows us to examine how well the methods could capture the *P* climatology but also the nonstationary changes of long-term *P*. Again, Scenario1 notably underestimated the climatology of observations (by 71% on average) (Figure 3 and Figure 4a), due to the use of MAE as loss function. In general, all other DL scenarios and QDM_BI provide

satisfactory results on capturing hourly *P* climatology. Scenario4 also slightly underestimated the climatology of Stage IV (12% on average, Figure 4a). This scenario only includes atmospheric covariates as model inputs without using the corrected *P* of MERRA-2, indicating the information from covariates only are not sufficient to estimate hourly *P*. The climatology of Scenario3, Scenario5 and Scenario6 appear well matching with Stage IV in space, better than QDM_BI. Relative differences of climatology averaged over the study area between estimated and Stage IV are 1.5%, 1.8% and 0.38% for Scenario3,

Scenario5 and Scenario6, respectively, while it is 2.5% for QDM_BI. Compared to Scenario3 and Scenario5, the Scenario2 underestimated the climatology particularly around the coastal area (Figure 3), which indicates the added value from multitask learning (Scenario3) and atmospheric covariates (Scenario5). Figure 4a shows that QDM_BI has a relative larger variance and its KGE value is lower than the ones for Scenario 2, Scenario3, Scenario5 and Scenario6. Note that all the DL estimates appears to be blurrier than Stage IV, similar as what has been found in previous studies (e.g., Ravuri et al., 2021),

while the QDM_BI estimates are even blurrier than the DL estimates.

[Insert Figure 3]

[Insert Figure 4]

### 3.4  Daily and Monthly *P* estimates

We aggregated the hourly *P* estimates into daily and monthly time scales to evaluate the performance of daily total *P*

and monthly mean of hourly *P*. Overall, the KGE values for the daily total *P* are considerably greater than those for the hourly *P* (Table 3), which suggests temporal aggregation denoised the hourly precipitation data, leading to considerably higher correlation coefficient (*r* in Table 3), mainly contributing to higher KGE. Similarly, The KGE value for Scenario1 is the lowest since it highly underestimated the mean of daily total *P* (lower β), overestimated the variability (higher γ), and the correlation *r* is also lower compared to the other scenarios. The Scenario5 and Scenario6 have relative higher KGE scores

than other DL scenarios and QDM_BI for daily total *P*. Daily total *P* from QDM_BI has a comparable KGE score with the DL models, while overestimated the variability (higher γ) compared to most of the DL scenarios.

Figure 5 shows the daily total *P* time series for each year during the testing period for the Stage IV, six DL scenarios and QDM_BI, averaged over the study area. The results show that the daily total *P* time series from the DL models closely matched with the daily total *P* time series from Stage IV except Scenario1. Again, Scenario1 highly underestimated daily

total *P* with the lowest KGE value, suggesting the difficulties of MAE in handling highly unbalance feature of *P*. The daily total *P* from all the other five DL scenarios are much close to Stage IV with larger KGE values (close to or larger than 0.9)



than QDM_BI. For these five DL scenarios (Scenario2 to Scenario6), Scenario5 and Scenario6 perform better than the others, indicating incorporating corrected $P$ and/or multitask learning further improved daily total $P$ estimates. The bias corrected and downscaled daily total $P$ from QDM_BI, however, highly overestimated the daily total $P$ of Stage IV for

almost all the large precipitation events, because the bias correction process for QDM_BI was executed individually at each grid cell and did not consider spatial dependencies and nonlinear relationships between covariates and observations, resulting in nonstable estimations (Wang & Tian, 2022).

[Insert Figure 5]

Table 3 also summarized the statistics of monthly mean of hourly $P$. The KGE values for monthly mean of hourly $P$

are greatly increased, higher than the daily total $P$. Except Scenario1, the KGE values for the monthly mean are very close to each other with Scenario4 slightly lower than others including QDM_BI. Monthly mean from QDM_BI had a relatively higher $\square$, indicating overestimations of variability. Figure 6 presents the monthly mean time series of hourly precipitation for each month during the test period for Stage IV, the six DL model and QDM_BI, averaged over the study area. Similar to daily total $P$ time series, the monthly mean $P$ from all the DL models closely matched with the monthly mean time series

from Stage IV (KGE value greater than 0.9) except Scenario1 which highly underestimated the observations. The Scenario6 has the highest KGE score, followed by Scenario3, Scenario5, Scenario2 and Scenario4, which are consistently better than the KGE score from QDM_BI. These results indicate that incorporating weighted loss function (Scenario2 to Scenario6 compared to Scenario1), multitask learning (Scenario3 and Scenario6) and atmospheric covariates (Scnenario4 to Scenario6) improved monthly mean estimation. Similarly, the monthly mean from QDM_BI estimates have relatively larger variability

than Stage IV, resulting in a lower KGE value.

[Insert Figure 6]

**3.5 Extremes**

Table 4 summarized the statistics of hourly $P$ at 99th percentile and the annual maximum wet spell. The results show that Scenario1 highly underestimated hourly $P$ at 99th percentile (lower $\beta$ than 1) and overestimated variability (higher $\gamma$ than

1), resulting in a negative KGE score, suggesting the inadequacy of using regular MAE loss function. Scenario2 has the highest KGE score with a higher correlation coefficient (higher $r$) than the other scenarios. This is probably because the number of trainable parameters for Scenario2 is the lowest, leading to a better regularization ability with limited data for extremes. The KGE values are similar for Scenario3, Scenario5 and Scenario6, and relatively higher for Scenario4, suggesting the importance of incorporating observation corrected $P$ from coarse resolution as an input. The benchmark

approach QDM_BI highly overestimated the variability of hourly $P$ at 99th percentile compared to Stage IV, resulting in a lower KGE values than most of the DL scenarios except Scenario1.





Figure 4b shows the boxplots of the relative difference between hourly $P$ estimates and Stage IV observations at the 99th percentile. On average, Scenario1 underestimated the 99th percentile hourly $P$ by over 60%, while other DL scenarios underestimated by about 20% with Scenario5 and Scenerio6 much closer to Stage IV. The 99th percentile estimated by QDM_BI has a much higher variance (as indicated by the distance between high 90% and low 10% bars in boxplot, as well as high γ in Table 4) compared to DL models, while has a lower mean difference (underestimated by about 10%) due to bias correction through explicit adjustment at each percentile. Figure 7 shows the spatial distribution of the hourly $P$ at 99th percentile for MERRA2, Stage IV, QDM_BI and six DL models. We can see that 99th percentile of MERRA-2 hourly $P$ greatly underestimated Stage IV by 40% (spatial average 2.9mm for MERRA2 versus 4.8mm for Stage IV). While the hourly $P$ at 99th percentile from QDM_BI (area average 4.3mm) appears to be close to Stage IV, its spatial variability looks very different from Stage IV, probably due to QDM_BI correcting biases on a grid point basis. The spatial average $P$ at 99th percentile for the six deep learning models is 1.7mm, 3.9mm, 4.0mm, 3.7mm, 4.2mm and 4.1mm for Scenario1 to Scenario6, respectively, indicating that increasing model complexity decreased hourly $P$ mean biases (i.e., β in Table 4) at 99th percentile.

[Insert Figure 7]

The DL models treated hourly spatial $P$ data independently and did not explicitly account for temporal dependence. However, the DL models could potentially well reduce temporal biases if spatial $P$ data for each hour can be well corrected and downscaled. The annual maximum wet spell is a widely used extreme index on evaluating temporal dependence (e.g., Maraun et al., 2015). The wetness threshold for calculating the annual maximum wet spell index was set to 0.1mm/h, which is commonly used for radar hourly data (e.g., Tao et al., 2016). Table 4 shows that Scenario2 and Scenario3 have relative higher KGE scores for the annual maximum wet spell extreme index than the other DL scenarios, suggesting the usefulness of more parsimonious models with weighted loss function but without including atmospheric covariates as additional inputs. Further incorporating multitask learning (Scenario3 and Scenario6), however, slightly decreased the model performance compared to no multitask learning scenarios (Scenario2 and Scenario5), probably due to the increased parameters and decreased regularization ability. While scenario1 has the lowest KGE score than the other DL scenarios, it is still much higher than QDM_BI which highly overestimated the mean of annual maximum wet spell for Stage IV observations (much higher β than 1). Boxplots in Figure 4c show the difference between model estimates and Stage IV observations for the annual maximum wet spell in hours during the test period. Scenario1 highly underestimated the annual maximum wet spell by about 10 hours. Scenario2 and Scenario3 have the lowest differences with Stage IV in terms of the mean and variance of the annual maximum wet spells. On average, Scenario4, Scenario 5 and Scenario6 overestimated the annual maximum wet spell by about 10 hours with Scenario4 and Scenario6 showing a relative larger variance. The benchmark approach QDM_BI has the largest difference (on average over 22 hours) and much larger variance compared to Stage IV, resulting in a negative




KGE score. This is probably because QDM_BI corrected biases on a grid basis, which failed to account for the spatial and temporal dependence.

Figure 8 shows an extreme event occurred from 19:00 to 20:00 on 29 August 2021 in Universal Time Coordinated (UTC) time zone when Hurricane Ida landed at the Louisiana State in the United States from MERRA2, Stage IV, QDM_BI and the six DL scenarios. We can see that MERRA2 highly underestimated this extreme event and did not capture detailed features of Stage IV. While QDM_BI estimates slightly enhanced the hourly $P$ values, it still failed to capture detailed feature. The Scenario1 to Scenario3 gradually enhanced hourly $P$, but these three models had difficulties to capture the

center of hurricane. By including atmospheric covariates, Scenario4 to Scenario6 roughly captured the center of hurricane and Scenario6 also reproduced the cyclones surrounding the center. These results suggest the importance of incorporating weighted loss function, multitask learning, and atmospheric covariates for bias correcting and downscaling specific extreme events.

[Insert Figure 8]

**3.6  *P* categories**

    Figure 9 shows that Scenario3 and Scenario6, the scenarios with multitask learning for bias correcting $P$ categories, have larger IoU values than QDM method particularly for the three categories with rain, indicating that the two DL models results well matched with the wet categories of the coarsened Stage IV observations, better than the QDM method. Furthermore, Scenario6 has relative larger IoU scores than Scenario3, indicating incorporating atmospheric covariates

improved classification accuracy. These results suggest that, with an auxiliary classification task, the Scenario3 and Scenario6 of DL model can well estimate the four categories of hourly $P$ during the testing period.

[Insert Figure 9]

**4    Discussion**

    This study explored customized DL for bias correcting and downscaling hourly $P$ through a set of experiments with or

without customized loss functions, multitask learning, and inputs from atmospheric covariates of precipitation. Scenario1, which used regular MAE as loss function, highly underestimated $P$ for all the temporal scales as well as extremes, showing the lowest performance. Since most of hourly $P$ are no rain, the regular loss function very likely leads the model to learn no rain events while neglecting rainy events. However, the scenarios with customized loss function with weighted MAE (Scenario2 to Scenario6) consistently showed much better performance than Scenario1. This result suggests that penalizing

more towards heavy $P$ on a grid basis makes the optimization algorithm focus more on the grids where rainfalls occurred and therefore inherently rebalance the hourly $P$ for model training.



The scenarios with multitask learning perform generally better than the other scenarios in terms of hourly climatology (Figure 4a), and daily and monthly assessments (Figure 5 and Figure 6). Multitask learning model with covariates can enhance extreme events and is the best model for application of bias correcting and downscaling *P* extreme events. Their
added values, however, are limited and performed worse than other scenarios without multitask learning (Scenario2 and Scenario5) in terms of extreme indices (see Figure 4b, 4c and Table 4). The reason for that is probably because adding multitask learning increased 30% trainable parameters with limited extreme data decreased the model regularization ability. Baño-Medina et al. (2020) designed a series of DL models with plain CNN architecture and different model complexity (i.e., increasing the number of model trainable parameters) to downscale daily ERA5 reanalysis dataset and found that increasing
model complexity make model performance worse particularly for extreme indices (98[th] percentile and annual maximum wet spell), which is consistent with our study.

Traditional methods (e.g., QDM_BI) mainly use coarse resolution *P* data as the only predictor for downscaling and bias correction, which cannot fully utilize nonlinear relationships between covariates and observations (Rasp & Lerch, 2018) during the bias correction and downscaling process. DL models with covariates as auxiliary variables, however, have
indicated success on improving model performance for postprocessing temperature and precipitation forecasts due to capability of learning nonlinear relationships between covariates and response variable automatically (W. Li et al., 2022; Rasp & Lerch, 2018). Scenario4 to Scenario6 incorporated physically relevant covariates of precipitation, with only Scenario4 excluding the coarse resolution *P*. The results indicate that incorporating auxiliary predictors of atmosphere circulations and moisture conditions can help improve *P* bias correcting and downscaling skill (see Figure 3 to Figure 8).
However, only using covariates without coarse resolution *P* (Scenario4) is not sufficient to well estimate hourly *P*, while using coarse resolution *P* as additional input (Scenario5 and Scenario6) shows improved performance. This result is consistent with a recent study focusing on CNN-based postprocessing of *P* forecasts from numerical weather prediction models, showing total precipitation itself is the most important predictor (W. Li et al. (2022).

Moreover, we compared the customized DL scenarios with a traditional method QDM_BI and found that all DL
experiments remarkably outperform QDM_BI in all the temporal scales as well as extremes. QDM_BI executed bias correction at each grid point without considering spatial dependencies and only used coarse resolution *P* as a predictor, and thus does not have the capability of capturing spatial features (e.g., detailed spatial features for the hurricane Ida in Figure 9) and accounting for the atmosphere and moisture covariates of precipitation. Furthermore, the proposed customized DL models are fully convolutional, and the trained models potentially can be easily used to estimate hourly *P* in other places
through transfer learning where high resolution data are not available [e.g., Stage IV radar coverage is limited in the western United States as a result of the scarcity of the radar network and blockage from the mountains (Nelson et al., 2016)]. The performance of transfer learning under various climate zones with different types of *P* events deserves a separate study. The



trained models also have the potential to generate high resolution hourly $P$ estimates beyond the time range covered by Stage IV radars (e.g., before 2002).

**5   Conclusions**

Various gridded precipitation ($P$) data at different spatiotemporal scales have been developed to address the limitations of ground-based $P$ observations. These gridded $P$ data products, however, suffer from systematic biases and spatial resolutions are mostly too coarse to be used in local scale studies. Many studies based on DL approaches have been conducted to bias correct and downscale coarse resolution $P$ data. However, it is still challenging for traditional approaches
as well as current DL approaches to capture small scale features especially for $P$ extremes due to the complexity of $P$ data (e.g., highly unbalanced and skewed) particularly at fine temporal scale (e.g., hourly). To address these challenges, this study developed customized DL models by incorporating customized loss functions, multitask learning and physical relevant atmospheric covariates. We designed a set of model scenarios to evaluate the added values of each component of the customized DL models. Our results show that customized loss functions greatly improved model performance compared to
the model scenario with regular loss function in all the temporal scales as well as extremes (on average improved by over 70% for climatology and over 50% at 99$^{th}$ percentile). The scenarios with multitask learning performed generally better than other scenarios on hourly climatology and aggregated time scales (daily and monthly), while the improvement is not as large as incorporating weighted loss function. While multitask learning greatly improved model performance on capturing detailed features of extreme events (e.g., hurricane Ida), the scenarios with multitask learning performed worse than other scenarios
in terms of extreme indices potentially due to the increased number of trainable parameters. The added value of incorporating atmospheric covariates is remarkable, likely because these scenarios took full advantages of nonlinear relationships between large-scale covariates, precipitation, and fine-scale observations.  The results also indicated that the role of coarse resolution $P$ as a predictor is very important for improving model performance despite the added values from the covariates. The DL scenarios with customized loss function and coarse resolution $P$ as the only predictor are the best
models at places where no covariate data are available. Moreover, all the DL scenarios with customized loss function performed much better in all the temporal scales as well as extremes than the benchmark approach QDM_BI, which is not able to account for spatial dependence and nonlinear relationships. These results highlight the advantages of the customized DL model compared with regular DL models as well as traditional approaches, which provides a promising tool to fundamentally improve precipitation bias correction and downscaling and better estimate $P$ at high resolutions.

**Code Availability Statement**

The code of regular and customized SRDRN models is available at: https://osf.io/whefu/ (DOI: 10.17605/OSF.IO/WHEFU).



**Data Availability Statement**

The MERRA2 product is accessible through the Goddard Earth Sciences Data Information Services Center (GES DISC;
http://disc.sci.gsfc.nasa.gov/mdisc/overview). The Stage IV radar precipitation data can be acquired via the National Center
for Atmospheric Research (NCAR) data portal (https://data.eol.ucar.edu/dataset/21.093).

**Author contributions**

FW: Methodology, Conceptualization, Software, Validation, Formal analysis, Investigation, Data Curation, Writing –
Original draft preparation, Visualization. DT: Conceptualization, Methodology, Software, Validation, Formal analysis,
Investigation, Writing- Original draft preparation, Supervision, Funding acquisition, Project administration. MC: Resources,
Writing – Review & Editing.

**Competing interests**

The contact author has declared that none of the authors has any competing interests.

**Acknowledgments**

This work is supported by in part by the NASA EPSCoR R3 program (No. NASA-AL-80NSSC21M0138), by the NSF
CAREER program (No. NSF-EAR-2144293), and by the NOAA RESTORE program (No. NOAA-NA19NOS4510194).

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



Table 1. Deep Learning (DL) Experimental Design

| Experimental Runs (Scenarios) | Input | Output | Loss |
|---|---|---|---|
| Scenario1 | hourly precipitation ($P$) | $P$ | MAE |
| Scenario2 | $P$ | $P$ | Weighted MAE |
| Scenario3 | $P$ | $P$ + categorical $P$ | Weighted MAE +$\lambda*$Weighted cross-entropy |
| Scenario4 | Covariates w/o $P$ | $P$ | Weighted MAE |
| Scenario5 | Covariates w/ $P$ | $P$ | Weighted MAE |
| Scenario6 | Covariates w/ $P$ | $P$ + categorical $P$ | Weighted MAE + $\lambda*$Weighted cross-entropy |




Table 2. Selected atmospheric covariates for DL downscaling and bias correction

| NO | Name | Description |
| --- | --- | --- |
| 1 | H250 | Geopotential height at 250 hPa |
| 2 | H500 | Geopotential height at 500 hPa |
| 3 | H850 | Geopotential height at 850 hPa |
| 4 | Q250 | Specific humidity at 250 hPa |
| 5 | Q500 | Specific humidity at 500 hPa |
| 6 | Q850 | Specific humidity at 850 hPa |
| 7 | T250 | Air temperature at 250 hPa |
| 8 | T500 | Air temperature at 500 hPa |
| 9 | T850 | Air temperature at 850 hPa |
| 10 | U250 | Eastward wind at 250 hPa |
| 11 | U500 | Eastward wind at 500 hPa |
| 12 | U850 | Eastward wind at 850 hPa |
| 13 | V250 | Northward wind at 250 hPa |
| 14 | V500 | Northward wind at 250 hPa |
| 15 | V850 | Northward wind at 250 hPa |
| 16 | OMEGA500 | Omega (vertical wind) at 500 hPa |
| 17 | SLP | Sea level pressure |
| 18 | T2M | 2-meter air temperature |



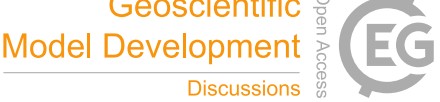

Table 3. Overall assessment for hourly, daily total, and monthly mean of hourly precipitation

| Temporal scales | Scenarios | KGE | r | β | γ | RMSE (mm) | MAE (mm) |
|---|---|---|---|---|---|---|---|
| Hourly precipitation | Scenario1 | -0.0584 | 0.267 | 0.288 | 1.28 | 1.20 | 0.189 |
| | Scenario2 | 0.218 | 0.297 | 0.958 | 0.660 | 1.25 | 0.258 |
| | Scenario3 | 0.203 | 0.278 | 1.02 | 0.664 | 1.28 | 0.269 |
| | Scenario4 | 0.250 | 0.331 | 0.883 | 0.682 | 1.21 | 0.240 |
| | Scenario5 | 0.283 | 0.358 | 1.02 | 0.682 | 1.22 | 0.248 |
| | Scenario6 | 0.262 | 0.356 | 1.00 | 0.639 | 1.20 | 0.247 |
| | QDM_BI | 0.248 | 0.332 | 1.02 | 1.35 | 1.36 | 0.256 |
| Daily precipitation | Scenario1 | 0.0935 | 0.615 | 0.288 | 1.409 | 10.19 | 3.54 |
| | Scenario2 | 0.644 | 0.685 | 0.958 | 0.840 | 8.76 | 3.42 |
| | Scenario3 | 0.626 | 0.675 | 1.02 | 0.815 | 8.94 | 3.54 |
| | Scenario4 | 0.618 | 0.642 | 0.883 | 0.935 | 9.37 | 3.55 |
| | Scenario5 | 0.688 | 0.701 | 1.02 | 0.914 | 8.89 | 3.40 |
| | Scenario6 | 0.668 | 0.701 | 1.00 | 0.855 | 8.65 | 3.34 |
| | QDM_BI | 0.644 | 0.689 | 1.02 | 1.17 | 10.50 | 3.42 |
| Monthly mean of hourly precipitation | Scenario1 | 0.0206 | 0.567 | 0.289 | 1.52 | 0.162 | 0.133 |
| | Scenario2 | 0.766 | 0.778 | 0.958 | 0.941 | 0.0721 | 0.0512 |
| | Scenario3 | 0.784 | 0.791 | 1.02 | 0.951 | 0.0713 | 0.0505 |
| | Scenario4 | 0.690 | 0.712 | 0.883 | 0.991 | 0.0835 | 0.0592 |
| | Scenario5 | 0.778 | 0.782 | 1.02 | 0.964 | 0.0734 | 0.0519 |
| | Scenario6 | 0.776 | 0.783 | 1.00 | 0.945 | 0.0719 | 0.0511 |
| | QDM_BI | 0.717 | 0.777 | 1.02 | 1.17 | 0.0850 | 0.0553 |





Table 4. Performance of extreme indices including hourly P at 99% percentile and annual maximum wet spell in hours.

| Extreme indices | Scenarios | KGE | r | β | γ | RMSE | MAE |
|---|---|---|---|---|---|---|---|
| 99th percentile (mm) | Scenario1 | -1.306 | 0.352 | 0.358 | 3.12 | 3.150 | 3.101 |
| | Scenario2 | 0.367 | 0.415 | 0.806 | 1.14 | 1.049 | 0.946 |
| | Scenario3 | 0.243 | 0.264 | 0.828 | 1.04 | 0.978 | 0.876 |
| | Scenario4 | 0.204 | 0.242 | 0.763 | 1.06 | 1.255 | 1.153 |
| | Scenario5 | 0.255 | 0.284 | 0.863 | 1.15 | 0.858 | 0.744 |
| | Scenario6 | 0.245 | 0.271 | 0.845 | 1.12 | 0.922 | 0.800 |
| | QDM_BI | 0.158 | 0.244 | 0.900 | 1.36 | 0.793 | 0.655 |
| Annual maximum wet spell (hours) | Scenario1 | 0.153 | 0.275 | 0.621 | 1.22 | 12.2 | 10.3 |
| | Scenario2 | 0.293 | 0.302 | 1.11 | 0.988 | 9.17 | 7.14 |
| | Scenario3 | 0.291 | 0.302 | 1.07 | 1.10 | 9.33 | 7.03 |
| | Scenario4 | 0.121 | 0.282 | 1.46 | 1.21 | 17.0 | 12.7 |
| | Scenario5 | 0.193 | 0.335 | 1.44 | 1.11 | 15.8 | 12.2 |
| | Scenario6 | 0.152 | 0.306 | 1.47 | 1.14 | 16.6 | 12.6 |
| | QDM_BI | -0.209 | 0.173 | 1.88 | 1.09 | 26.6 | 22.2 |






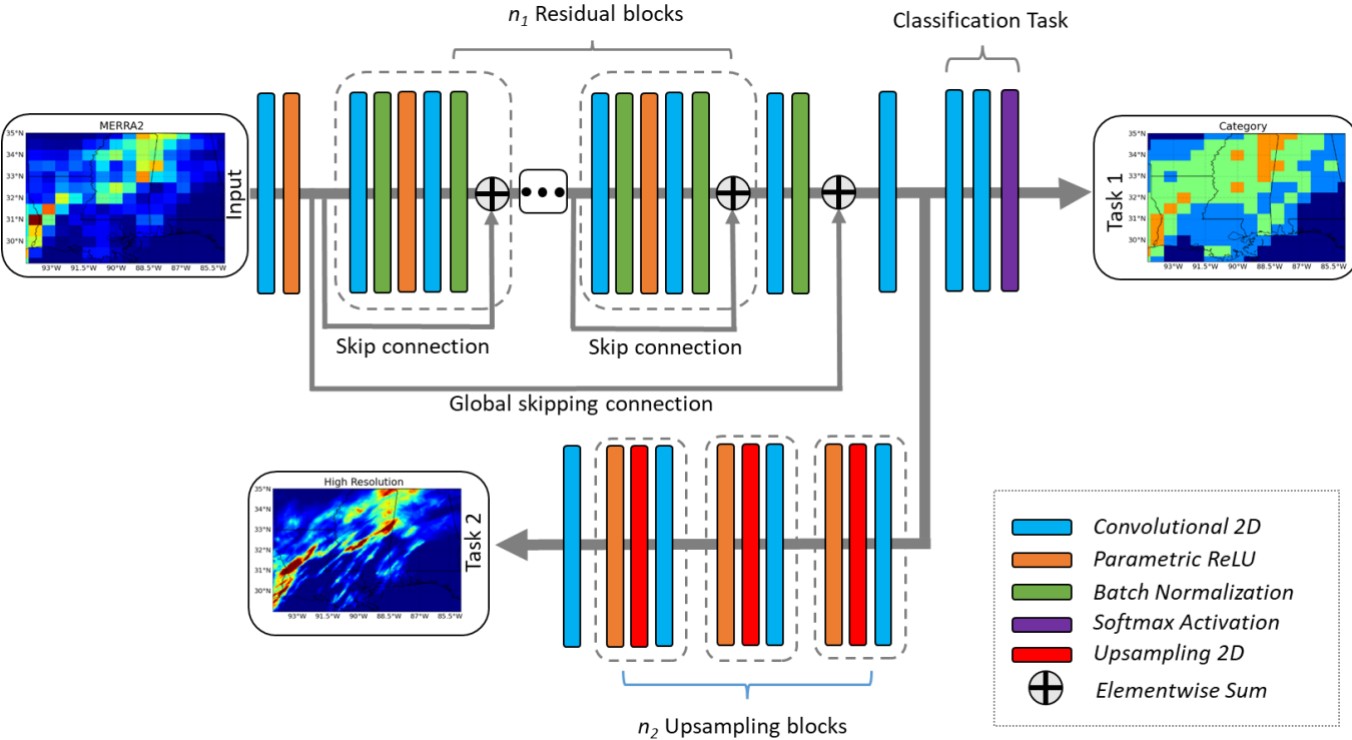

Figure 1. The modified SRDRN architecture with multitask learning, which included classification of P categories as an auxiliary task (Task 1) in addition to downscaling and bias correcting P values (Task 2).


Figure 2. Climatology of hourly precipitation from MERRA2 and Stage IV during the training period (2002 to 2015; first row) and their differences (second row) between the testing (2019 to 2021) and training periods.



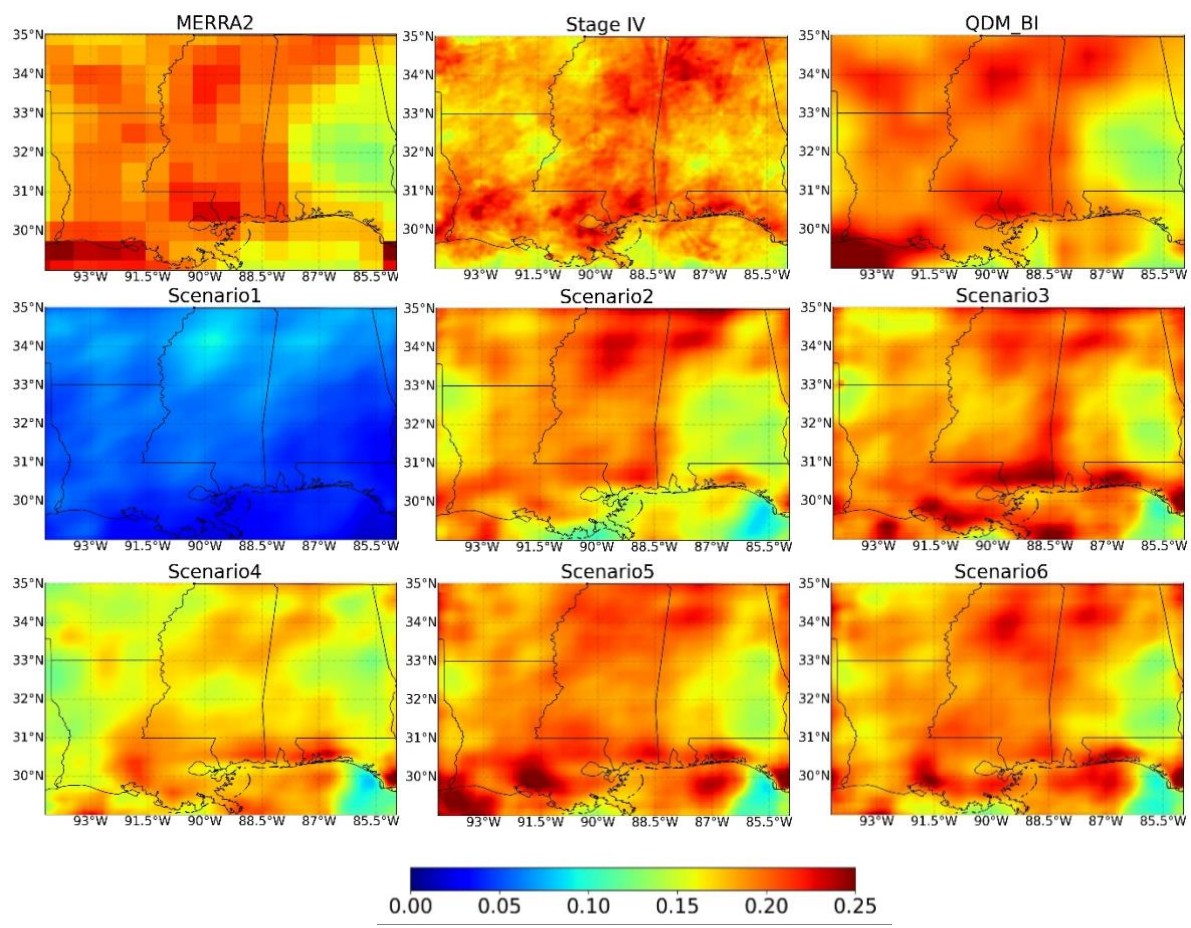

Figure 3. Hourly precipitation climatology during testing period (2019 to 2021), which includes MERRA2, Stage IV, QDM_BI and six DL experimental runs (Scenario1 to Scenario6).



Figure 4. Boxplots showing hourly precipitation estimates minus Stage IV observations based on, (a) climatology, (b) extreme at 99% percentile, and (c) annual maximum wet spell in hours during testing period (2019 to 2021). Precipitation estimates are produced from the QDM_BI approach and 6 DL experimental runs (Scenario1 to Scenario6).



Figure 5. Daily total precipitation during the testing period (2019 to 2021) from Stage IV, QDM_BI and 6 DL experimental
runs (Scenario1 to Scenario6).



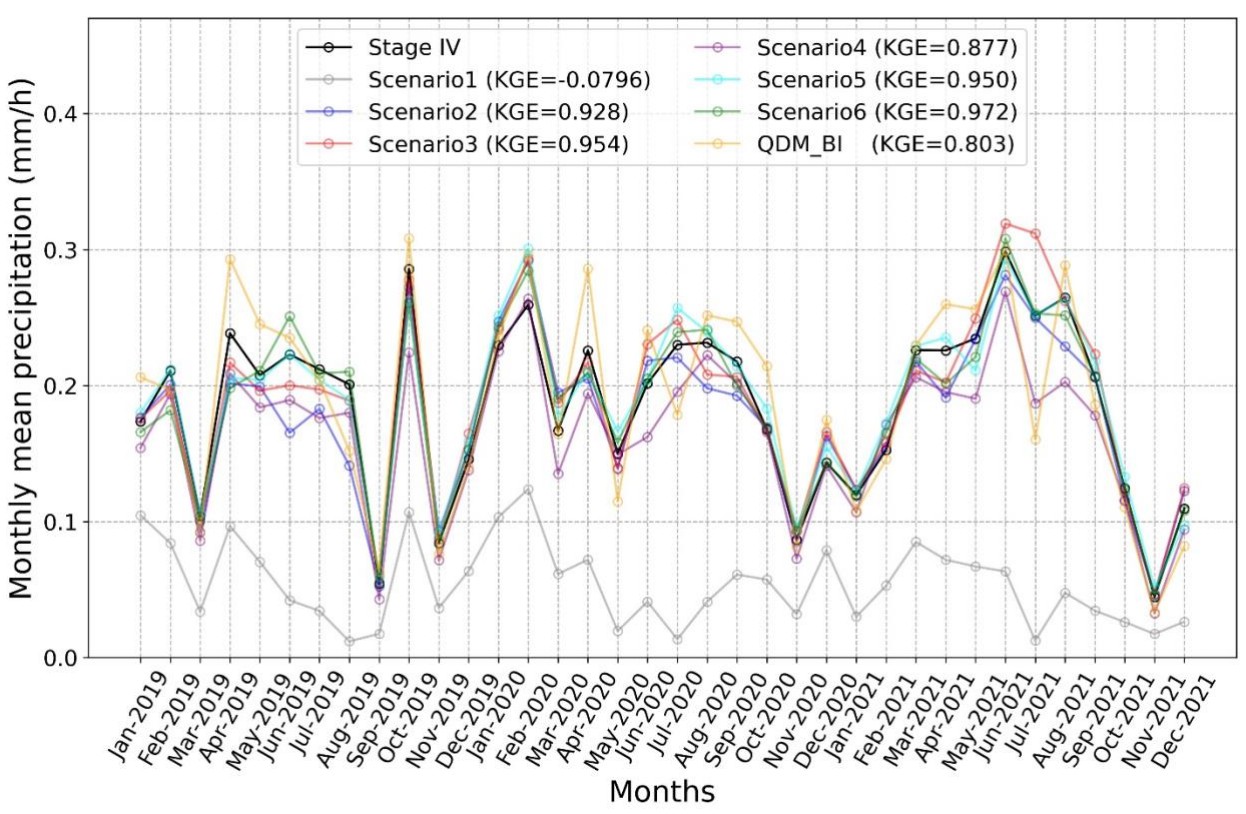

Figure 6. Monthly mean of hourly precipitation time series during the testing period (2019 to 2021) from Stage IV, QDM_BI and 6 DL experimental runs (Scenario1 to Scenario6).




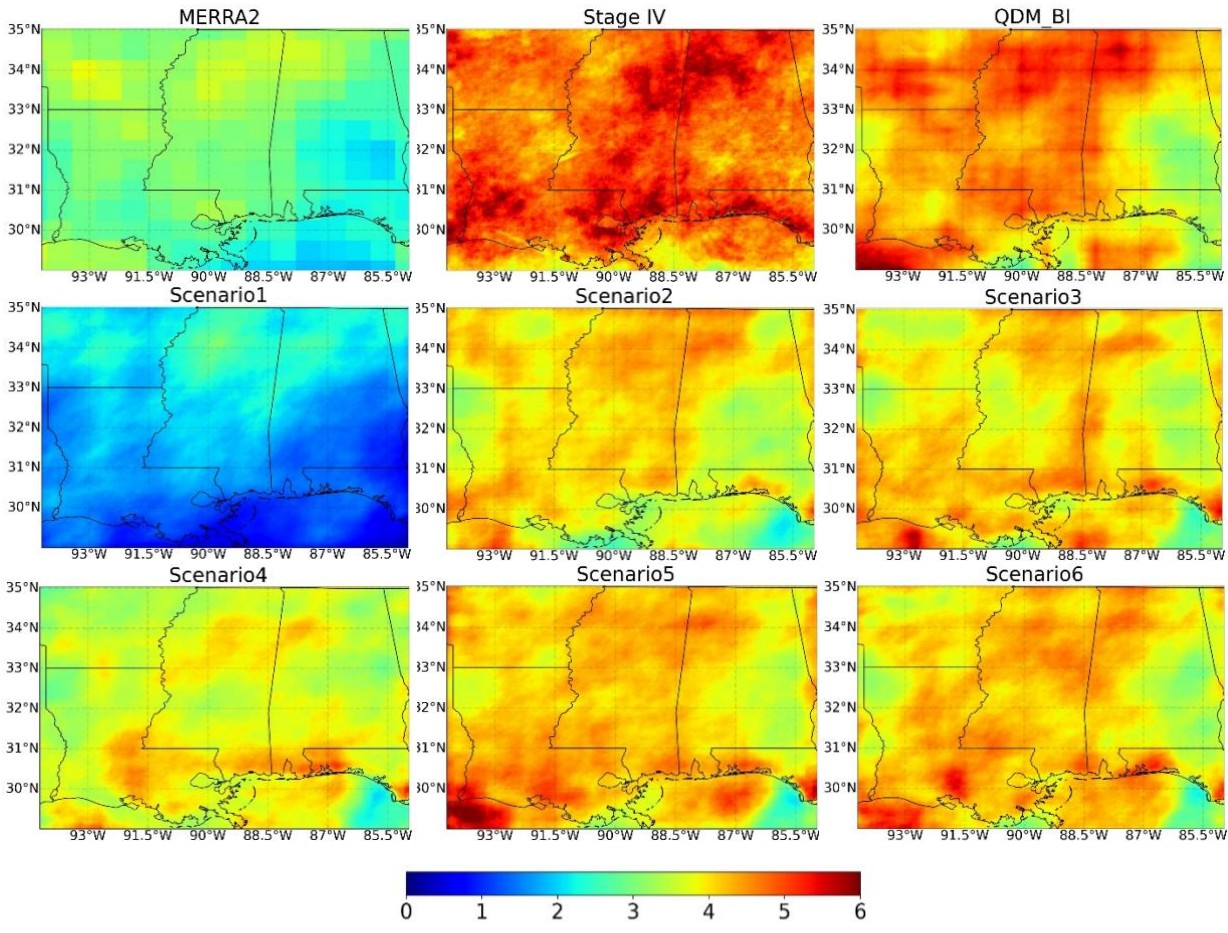

Figure 7. Spatial map of hourly precipitation extreme at 99th percentile from raw MERRA2, Stage IV, QDM_BI and 6 DL experimental runs (Scenario1 to Scenario6).

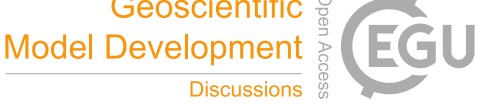



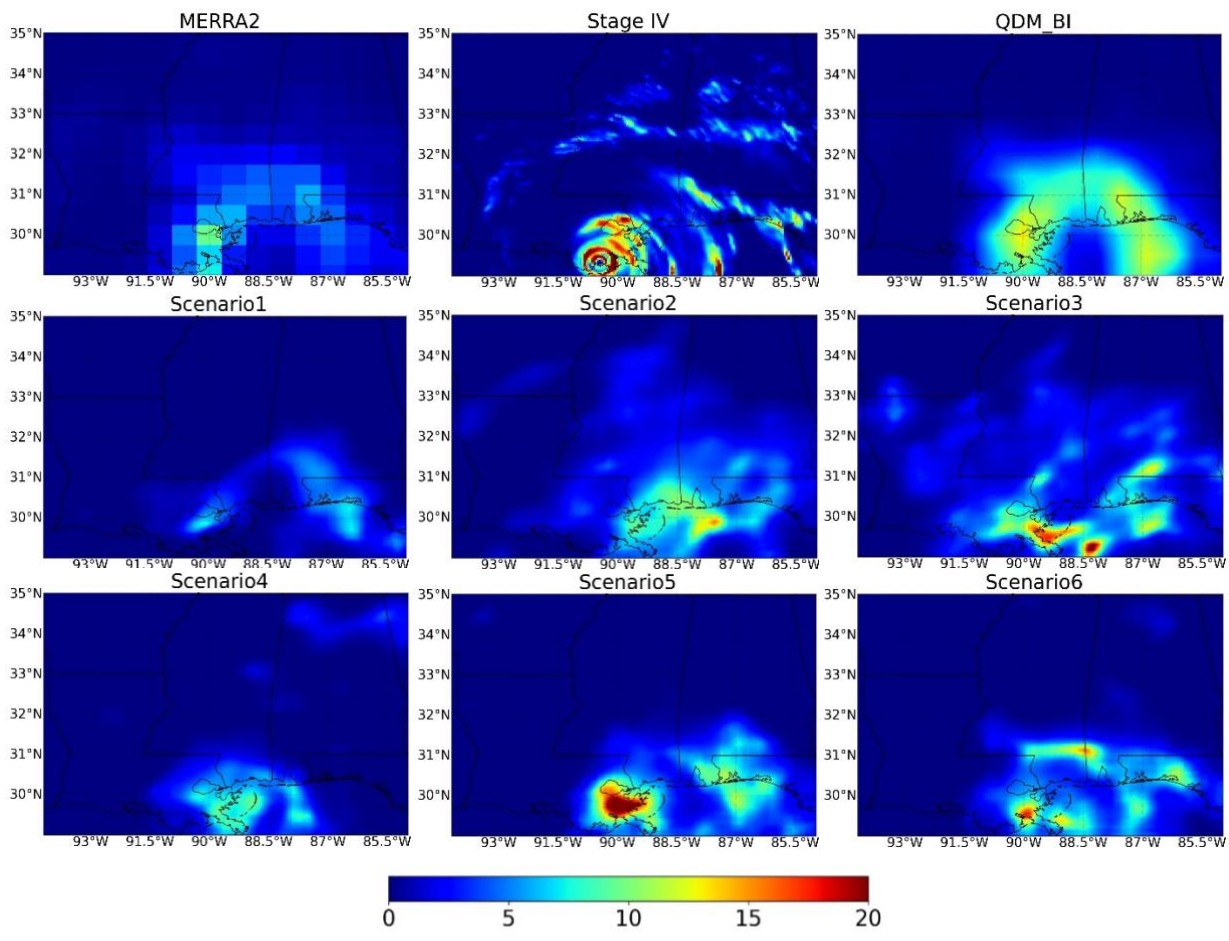


Figure 8. Hourly precipitation from 19:00 to 20:00 on 29 August 2021 in UTC time zone when Hurricane Ida landed in Louisiana, including raw MERRA2, Stage IV, QDM_BI and six DL experimental runs (Scenario1 to Scenario6).





| Categories | MERRA2 | QDM | Scenario3 | Scenario6 |
|---|---|---|---|---|
| 0-0.1mm | 80.54 | 88.10 | 81.00 | 86.44 |
| 0.1-2.5mm | 27.10 | 23.60 | 25.93 | 27.91 |
| 2.5-10mm | 14.94 | 15.30 | 19.63 | 19.91 |
| >10mm | 4.32 | 7.12 | 8.15 | 11.07 |

Figure 9. Heat map showing the Intersection over Union (IoU) comparing coarsened Stage IV with raw MERRA2, QDM, two deep learning experiment runs with classification task (Scenario3 and Scenario6)