# Peer review of "Customized Deep Learning for Precipitation Bias Correction and Downscaling"

_Geoscientific Model Development, 2022_

## Referee Comment (RC6)

**Review of "Customized Deep Learning for Precipitation Bias Correction and Downscaling" by Wang *et al.* (gmd-2022-213)**

The article proposes some improvements to the authors model for downscaling precipitation presented in Wang *et al*, (2021) where they use the loss function MSE instead of MAE.

The three proposed improvements are:

- using a weighted MAE as the loss instead of the MAE,

- using a second loss function on a quantized version of the upscaled Stage IV data

- and including other coarse grained predictors.

They evaluate these improvements in the task of downscaling hourly precipitation from the coarse grained MERRA2 ( $50\text{km}^2$) to the fine grained Stage IV ( $4\text{km}^2$) in a rectangle coastal area of the Gulf of Mexico covering the states of Alabama, Mississippi and Louisiana. They evaluate the performance of models by comparing the KGE-score on different aggregations as well as extreme events. The authors conclude that all three of their proposed improvements are helpful. In two marginal notes, the authors evaluate whether coarse-grained predictors make precipitation redundant as an input and whether model performance is related to its complexity. While they state the first to be negative, they state the second to be true.

The problem of downscaling precipitation is relevant and tailoring proven deep learning methods to this problem is a valuable contribution. However, the presented study has severe issues that considerably weaken its interpretation and the possible impact of the study considerably. Further, parts of the manuscripts need major updates.

This article requires **major revisions** before publication.

**General comments**

**Study**

Unfortunately, the results presented in Tables 3 and 4 are not enough to estimate the usefulness of the three proposed improvements. The differences between the "Scenarios" 2 to 6 are marginal and the order differs a lot between tasks and metrics. This is especially critical, since the chosen method (a deep neural network) is inherently stochastic and hence, differences between different "Scenarios" might be due to this stochastisity. This stochastisity is further increased by the special training method that the authors use. Instead of presenting each time step once in each epoch, they present random 1897 independent sampled batches of 64 random time-steps (which should be mentioned in the manuscript). To distinguish between these random effects and the effect of the proposed improvements it would be necessary to run the models multiple time and assess the significance of the differences between results.

Further, to evaluate the three different improvements independently it would be interesting to test and compare all eight possible combinations. The paper, unfortunately, only reports results on five of the eight combinations. Having the three missing combinations (standard MAE + categorical Loss, standard MAE + covariates and standard MAE + categorical Loss + covariates) will help immensely in disentangling the effects of the individual improvements.

The only difference that is apparent without the need of a test of significance is the difference between the "Scenarios" that use the weighted MAE and the "Scenario" that use the standard MAE. However, it is unclear, why the authors choose to modify the baseline (Wang *et al.*, 2021) to use a MAE instead of a MSE. In fact, to understand the performance of the proposed model in comparison to the state of the art, a comparison to the original baseline would be very helpful. Especially since

the baseline reached a KGE of 0.951 on a slightly different task, which indicates that it might be very competitive. Especially to support claims like "These results highlight the advantages of the customized DL model compared with regular DL models as well as traditional approaches, which provides a promising tool to fundamentally improve precipitation bias correction and downscaling and better estimate P at high resolution."[lines 502 to 505]

Since the central result of the paper seems to be that MAE is not a suitable loss for downsampling precipitation, the paper should include a discussion on why someone might consider this to be a sensible idea in the first place, which will amplify the impact of the result. However, at this stage, the motivation for this change in the baseline is unclear from the paper and (to the best of my knowledge) it was not suggested to use MAE in the literature (at least not in the related work presented in the paper).

Finally, the two side nodes of whether the coarse grained precipitation can be excluded as a predictor ("Scenario4") and whether larger models are overfitting in this specific example do not fit naturally in this study and distract from the main point of the study. Since both of them cannot be answered significantly from the results, I recommend to exclude them to further the readability.

**Manuscript**

Many of the above mentioned comments have implications on the manuscript. For example the discussion is quite long and discusses many aspects that are not reflected in the experimental results in any significant way (lines 307-314, 330-340, 349 - 351, 357-362, 367-375, 380-387, 388-392, 405-419, 422-428, 447-479). I recommend to focus the discussion of results mainly on significant results to not "over-interpret" the results and, consequently, "over-claim".

Further, many of the figures require more work. Figure 1 should maybe reference the very similar figure in (Wang *et al.*, 2021). Many figures have incomplete or no colorbars. Figure 5 and 6 are hard to read and I recommend to exclude them. The interpretation of Figure 9 is unclear.

Additionally, it would be helpful if the motivation for each of the three individual contributions is clearly stated in the beginning of the paper.

Further, the structure of the paper is unclear, for example "Data and methodology" is combined into one section, but is immediately split into two parts, which are data and methodology in 2.1, 2.2. It would be helpful to my understanding of the manuscript to restructure the work.

The notation of different models as "Scenarios" is confusing.

Often the choice of references is confusing. For example Li *et al.* (2021) is cited for IoU even though the paper includes no information on IoU that is not also included in this manuscript. This is just an representative example for other cases.

Finally, the interpretation of the KGE, more specifically the interpretation of $\beta$ and $\gamma$ is surprising. The authors, for example, state that "Scenario1" "highly overestimated the variability"[line 306] however, if we calculate

$$\frac{\sigma_s}{\sigma_o} = 0.37$$

indicating, that the variance is actually under estimated.

**Summary**

In summary I believe that the study aims to close a relevant research gap. Further, the proposed method of testing different models with combinations of different improvements is effective. By repeating the experiments to reach significant results, comparing the results to a state-of-the-art baseline and adding more explanation on the motivation of the proposed changes, the paper will be a valuable contribution.

---

## Author Response (AR1)

**Response to RC1**

For local scale studies, current precipitation datasets are crucial for bias correction and downscaling. By using customized loss functions et al. to bias correct and downscale hourly precipitation data, the authors developed a customized DL model based on the SRDRN architecture. This model provides better precipitation estimates at fine spatial and temporal resolutions. However, there are still some problems need to be solved:

1. Page 5, Line 130. Although the authors acknowledge that SRDRN performs better than conventional methods, they do not discuss its benefits over other types of deep learning models, and it is unclear why SRDRN was selected.

*Response: Thank you for your comments. We have included the following explanations in Section 3.1.1 of the revised manuscript: "Furthermore, it has been proved that the SRDRN is capable of capturing much finer features than shallow plain CNN architecture (Wang et al. 2021). Comparing with the popular U-Net architecture (Sha et al. 2020b; Sun and Tang 2020), the SRDRN directly extracts feature on the coarse resolution input, and thus can potentially decrease computational and memory complexity."*

2. Page 6, Line 158. The authors mention that the precipitation classification task and the correction and downscaling tasks are highly relevant. Is this an empirical or theoretically based judgment?

*Response: Thank you for your comments. Studies have shown that a multitask DL model could learn to better represent the shared physical processes and better predict the variable of interest (e.g., Sadler et al., 2022). Since P categories and actual values are highly relevant, we expect the added classification task can improve the DL model for bias correcting and downscaling P. We have modified the statement as follows in the Section 3.1.2 of the revised manuscript: "Studies indicated that a multitask DL model could learn to better represent the shared physical processes and better predict the variable that we are interested in (e.g., Sadler et al., 2022). Since P categories and actual values are highly relevant, adding a classification task can potentially improve the DL model for bias correcting and downscaling P."*

3. Page 7, Line 191. How does weighted cross-entropy as a loss function penalize the heavy rain category more?

*Response: Thank you for your comments. Since the weights for categories with rain are relatively larger than no rain category, the weighted cross-entropy loss is relatively large when there are discrepancies between true and predicted categories with rain, resulting in guiding the training towards to decreasing these differences with larger weights and thus better handling class-imbalance issue. We have included the following explanations at the end of Section 3.1.3 of the revised manuscript: "Since the weights for categories with rain are relatively larger than no rain category, the loss $L_{weighted\ Cross-entropy}$ is relatively large when there are discrepancies between true and predicted categories with rain, resulting in*

*guiding the training process towards to decreasing these differences with larger weights and thus better handling class-imbalance issue.”*

4.  Page 11,Line 296. The author should explain the exact meaning of TP, FP and FN, similar to TP (prediction=1, truth=1).

*Response: Thank you for your comments. We have included the following explanations in Section 3.3 of the revised manuscript: “where TP represents true positives (prediction=1, truth=1), FP represents false positives (prediction=1, truth=0) and FN represents false negatives (prediction=0, truth=1). Taking the heavy rain category as an example, TP is an outcome where the model correctly predicts the heavy rain class; FP is an outcome where the model predicts it is a heavy rain class, but the true label is not a heavy rain class; FN is an outcome where the model predicts it is not a heavy rain category, but the true label is a heavy rain class.”*

5.  Section 3. No subsection 3.1, layout error or omission?

*Response: Thank you for pointing that out. We have fixed it.*

6.  Table 2. The authors need to have described the units of each variable in the table.

*Response: Thank you for pointing that out. We have add a unit column in Table 2.*

7.  Figure 2,3,7 and 8. The authors should label the units of the physical quantities represented in the figure next to the legend.

*Response: Thank you for pointing it out. We have added units in the caption of each figure.*

**Response to RC2**

In this paper, the author customized a deep learning model to bias correct and downscale hourly precipitation data over the coastal region of the Gulf of Mexico. The study considered six different scenarios with different initial parameters to assess the added value in the model performance reproducing the local features. Data from MERRA2 reanalysis was used as predictor and data from stage IV radar as predictand. The results were assessed using the statistical matrices at different time scales. The findings in this paper are very interesting. In general terms this paper falls within the scope of this journal, the figures and tables are well organized, and the results are properly discussed. However, a few minor comments must be addressed:

*Specific comments:*

1. In the upsampling layers the author used the nearest neighbor interpolation method, this method played a fundamental role in increasing the spatial resolution of the results from the residual blocks, the author needs to justify why he chose this particular interpolation method.

*Response: Thank you for your comments. The nearest neighbor interpolation method is the defaulted interpolation method in the Upsampling2D layer in Keras API and effects of other interpolaton methods were not explored in this study. We have included the following explanations in Section 3.1.1 of the revised manuscript: "The defaulted nearest neighbor interpolation was chosen in the upsampling layers to increase spatial resolution and effects of different interpolation methods were not explored in this study."*

2. During the training phase, some scenarios consider atmospheric covariates of precipitation from MERRA as predictors, what about the predictand? Did the author use the same variables from stage IV? Also, it's not clear how the author aggregated those covariates to generate precipitation.

*Response: Thank you for your comments. Scenario4 to 6 considered atmospheric covariates from MERRA2 as predictors and the predictand is stage IV radar precipitation. We did not use any other variables from stage IV and did not aggregate any covariates. The selected covariates are physically relevent to precipitation and studies have shown those covariates are helpful for estimating precipitation. For example, Li et al. (2022) used CNN-based approach to postprocess numerical weather prediction model output and found that the use of auxiliary predictors greatly improved model performance compared with using raw precipitation data as the only predictor. See the detailed information in Section 1 of the revised manuscript. We also explained the motivations of including covariates in Section 3.1.4 of the revised manuscript as follows: "As described in Section 1, studies have indicated that including atmospheric covariates is helpful for estimating precipitation (e.g., Baño-Medina et al., 2020; Li et al., 2022; Rasp and Lerch, 2018)." The general form that using those covariates to estimate precipitation is comparable to a classic multiple linear regression problem (multiple variables as predictors and only one preditand). We have included the following explanations in Section 3.1.4: "Comparable to a classic multiple linear regression problem, covariates are multivariable predictors and hourly precipitation is the only dependent variable."*

3. In the results section, the first subtitle "Overall agreement" should start from 3.1.

*Response: Thank you for pointing that out. We have fixed it.*

4. The author could elaborate more on why the 6 scenarios resulted in a low correlation ratio at hourly time scale.

*Response: Thank you for your comments. We have included the following explanation in Section 4.1 of the updated manuscript: "The reason is because the correlation component $r$ was estimated based on all the hour to hour P data, while the other two components (i.e., $\beta$ and $\gamma$) were calculated based on long-term climatological P statistics and were relatively easier to estimate (Beck et al., 2019b)."*

5. It is suggested to add the definition of r, β, and Υ in the caption of Tables 3 and 4.

*Response: Thank you for your comments. We have added their definition in the caption of tables 3 and 4.*

6. Figures 2 and 3: the unit beside the colorbar is missing, is it (mm/hour)?

*Response: Thank you for pointing it out. We have added units in the caption of each figure. Yes, it is mm/h.*

7. Page 12: "Monthly mean from QDM_BI had a relatively higher…" something is missing in this sentence.

*Response: Thank you for pointing it out. Monthly mean from QDM_BI had a relatively higher $\gamma$ and $\gamma$ is missing. We have fixed it in the revised manuscript.*

**Response to RC3**

The paper entitled "Customized Deep Learning for Precipitation Bias Correction and Downscaling" developed a customized deep learning model for precipitation downscaling and tested the advantages of incorporating a weighted loss function, multitask learning, and accounting for physically relevant covariates. The manuscript is generally well structured and clearly presented. I have a few minor comments regarding the explanations of the training processes.

1. Line 88: (Chen, 2020)

*Response: Thank you for pointing it out. We have fixed it.*

2. Section 2.2.4 How many training steps (iterations) do you have for each scenario? It would be nice to see the learning curve for each scenario.

*Response: Thank you for your comments. The total iterations for each scenario is about $2.5\text{x}10^5$ and the learning curves for the 6 scenarios are ploted in Figure S1 in the Supplement document. We have added the information in Section 4.1 of the updated manuscript.*

3. Line 172: Please explain why you specifically developed the weighted mean absolute error (MAE) loss function rather than using the regular MAE.

*Response: Thank you for your comments. We have added the following explanation in Section 3.1.3 of the updated manuscript: "Precipitation data is highly skewed and unbalanced especially at hourly time scale, which could cause deep learning algorithm to focus more on no rain events and ignore heavy rain events if using regular loss functions."*

4. Line 211: How are these covariates chosen? I wonder if the authors have tested if all covariates are necessary for precipitation downscaling or if other covariates are necessary (for example, convective available potential energy).

*Response: Thank you for your comments. These covariates are chosen based on precipitation formation theory (cloud mass movements and themodynamics) and other studies on estimating precipitation. We have included the following explanation in Section 3.1.4 of the updated manuscript "We chose these variables based on precipitation formation theory (cloud mass movements and thermodynamics) and other studies on estimating precipitation as listed above." We did not tested whether all covariates or other covariates are necessary in this study. We have made a note in the discussion section 5 of the updated manuscript as follows: "Note that we did not explore the importance rank among these covariates in improving the model performance in this study, which could be a potential avenue for future work."*

5. Line 240: Can you show the data distribution during the training period (2002-2015) and the test period (2019-2021)? It is hard to tell if the methods have extrapolation capability without seeing the differences in precipitation distribution. Are the increases in annual precipitations caused by a systematic increase each timestep or an increase in extreme precipitations?

*Response: Thank you for your comments. Since there are too many no rain hours, the data distributions during training and testing periods are hard to disgintuish. Therefore, we calculated the probability for each precipitation bin with a range of 0.5 mm/h during the training and testing periods and calculated their difference of probability for each bin (see Table S1 in the Supplement). The table suggests that the climatology differences between testing and training datasets (Figure 2) is because there is higher percengage of rains greater than 0.5 mm/h in the testing data, while there is a higher percentage of no rain or drizzling (0 to 0.5 mm/hr) in the training dataset. We have added the following explanation in the Section 3.1.4 of the updated manuscript: "Wetter conditions are observed in most of the study area in the testing period (average 0.03 mm/h) than the training period, which are due to a higher percentage of rains (with values greater than 0.5mm/h) during the testing period than during the training period based on analyzing the Stage IV data (Table S1 in Supplement)."*

*We train the model using hourly data and evaluate that at different scales including hourly, daily, and monthly scales. The increase in skill in aggregated time scale (e.g., daily and monthly) is due to the smoothing of the hourly data. We did not aggregate the data into annual time scale, but it is expected that the skill for annual time scale would be better than monthly scale due to further smoothing.*

6. Line 244: Briefly describe how QDM is applied in this study

*Response: Thank you for your comments. We have included more explanations in Section 3.2 of the updated manuscript as follows "The QDM method corrects systematic biases at each grid cell in quantiles of a modelled series with respect to observed values. Compared to the regular quantile mapping method (Panofsky and Brier, 1968; Thrasher et al., 2012; Wood et al., 2002), QDM also applies a relative difference between historical and future climate data (here, training and testing periods),  thus it is capable of preserving trend of the future climate (Cannon et al., 2015), which is critical for this study since there are substantial differences between the precipitation during the training (2002 to 2015) and testing (2019 to 2021) periods (see Figure 2)".*

7. Line 302: There is no section 3.1

*Response: Thank you for pointing it out. We have fixed it.*

**Response to RC4**

The article is well written and easy to understand. I have the following comments for the authors to consider.

1. P2. L55. Single image super-resolution approaches have attracted much attention for precipitation downscaling. In Vandal et al. 2019 cited here, the authors first coarsened a precip dataset and then tried to recover the original fine scale precip data. While that exercise had pedagogical meanings, it is not exactly very practical. Sun and Tang (2020) focused on a 3-deg study area in Texas and demonstrated downscaling and bias correction using multiple coarse-resolution satellite data and Attention GAN superresolution. That work is especially relevant to this study, as it focused local and fine features for an area near Gulf of Mexico. It can be added to the list of references here.

*Response: Thank you for your comments. We have added the reference of Sun and Tang (2020) in the introduction section of the updated manuscript.*

2. MERRA2 is reanalysis data. I wonder if near real time data from, e.g., GPM, should be used to demonstrate operational aspect of this algorithm.

*Response: Thank you for your comments. MERRA2 is a state-of-the-art global reanalysis product and it incorporates new satellite observations through data assimilation and benefits from advances in the GEOS-5, which also includes relevant atmospheric variables that can assist in estimating precipitation at fine scale as indicated by Scenario4 to Scenario6 in this study. Besides climate reanalysis, this algorithm can be potentially integrated with precipitation data from the Global Precipitation Measurement (GPM) mission to generate more accurate operational precipitation data at finer resolution. We have added the following information in the discussion section 5 of the updated manuscript "While this study explored bias correcting and downscaling hourly precipitation from climate reanalysis data, this algorithm with customized loss function can be potentially integrated with precipitation data from the Global Precipitation Measurement (GPM) mission to generate more accurate operational precipitation data at finer resolution."*

3. Most deep learning studies adopt some baseline. The authors may want to show a comparison to baseline. A meaningful baseline here can be bilinear interpolation.

*Response: Thank you for your comments. We compared the deep learning models with Quantile delta mapping with bilinear interpolation (QDM_BI) as the baseline approach. QDM was applied at coarse resolution to correct biases between MERRA2 and Stage IV precipitation and then used bilinear interpolation (BI) to increase the resolution. Please see detailed description of QDM_BI in Section 3.2 of the updated manuscript.*

4. P32. Figure 3. The downscaled maps seem to be smooth and lacking many details shown in Stage IV. Here the authors mainly considered climate covariates from MERRA2. I think adding static covariates such as DEM may help to resolve some local details.

*Response: Thank you for your comments. We agree with you that adding static variables could be helpful for resolving local details. We include a discussion about the potential room for improving the performance by incorporating static variables as follows "Furthermore, static variables, such as elevations, long term climatology (Sha et al., 2020a), soil texture and land cover, could be helpful for resolving local details. However, our study region has little topographic variations and therefore including elevation data cannot add any additional information to the model."*

References

Sha, Y., Gagne II, D. J., West, G., and Stull, R.: Deep-learning-based gridded downscaling of surface meteorological variables in complex terrain. Part II: Daily precipitation, Journal of Applied Meteorology and Climatology, 59, 2075-2092, 2020a.

Sun, A. Y., & Tang, G. (2020). Downscaling satellite and reanalysis precipitation products using attention-based deep convolutional neural nets. Frontiers in Water, 2, 536743.

**Response to RC5**

The paper presents a customized deep learning methodology that incorporates customized loss functions, multitask learning, and physically relevant covariates for bias correction and downscaling of precipitation data, starting from the hourly time scale. This is a very difficult task, due to the complexity of the precipitation characteristics. Improvements in the generation of temporal and spatial high-resolution data for practical applications and for scientific tasks, as validation of very high resolution nonhydrostatic atmospheric models. The bias correction procedure reflects one of the weakness of this kind of methodology: it relays on the estimation of climate trends based on the differences between the training and test periods. These differences could be related to low frequency climate variability and the bias correction procedure can be influenced for this variability.

The methodology is evaluated using six different scenarios to systematically evaluate the added values of weighted loss functions, multitask learning, and atmospheric covariates and compare its performance to the regular deep learning and statistical approaches, pointing to the usefulness of considering physical relationships in choosing the atmospheric covariates. This fact points to the necessity of the consideration of physical variables related to precipitation relationships for a better bias correction and downscaling.

The authors show that using atmospheric covariates also helps to improve the performance of the methodology for capturing extreme events. One important result is the ability that customized deep learning to improve downscaling and bias correction of precipitation estimates of gridded precipitation datasets with respect to more standard deep learning and statistical methods.

I consider this paper an important step forward in the task of the improvement of bias correction and downscaling of gridded data. It is especially important for the downscaling and bias correction of climate data generated for climate change impact studies, as there is a strong need in a good bias correction (taking caution about the low frequency variability problem) and downscaling for impact studies. The paper deserves to be published after some corrections

**Some questions:**

1. Will the use of a different interpolation method in the upsampling layers would influence the downscaling? Which is the limit for the downscaling factor?

*Response: Thank you for your comments. We did not explore other interpolation method in the upsampling layers except the default nearest neighbor method, while using different approaches may impact the result. We added an explanation in Section 3.1.1 as follows: "effects of different interpolation methods were not explored in this study". The downscaling factor in this study is 12. In our previous study based on synthetic experiments (Wang et al. 2021), we indicated that the downscaling factor of 24 still worked better than traditional statistical downscaling method.*

2. Which criteria do you use to choose the atmospheric covariates? The fact that despite the use of covariates still is not enough and you heavily need the precipitation for training the model reflects the necessity of considering other covariates?

*Response: Thank you for your comments. The covariates are chosen based on precipitation formation theory as well as findings from studies on estimating precipitation as listed in Section 3.1.4 of the revised manuscript. We have added the following explanation in Section 3.1.4 of the updated manuscript "We chose these variables based on precipitation formation theory (cloud mass movements and thermodynamics) as well as findings from previous studies as indicated above." We did not explore the importance among those covariates in this study and we have made a note in the discussion section 5 of the updated manuscript as follows: "Note that we did not evaluate the importance ranking among these covariates in improving the model performance in this study, which can be a potential avenue for future work."*

3. In Baño-Medina et al(2022) deep learning was used for downscaling the EUROCORDEX CMIP5 simulations. Could your methodology be used for a similar task?

*Response: Yes, the regular SRDRN has been investigated to downscale daily temperature from CMIP6 GCM outputs in the historical run (Wang et al., 2022). The SRDRN architecture can be further customized to downscale different gridded precipitation including downscaling precipitation from GCM projections, which can be a future study. We have added this information in the discussion section 5 of the updated manuscript.*

**Manuscript**

1. Line 27: correct "experiencing" with "experience"

*Response: Thank you for your comments. We have fixed it.*

2. Line 36 correct "data" with "datasets"

*Response: Thank you for your comments. We have fixed it.*

3. Line 325. It also can be related to natural climate variability.

*Response: Thank you for your comments. We have added "climate variability" in Section 4.2 of the updated manuscript.*

4. Line 401. Please, elaborate on this "The DL models treated hourly spatial P data independently and did not explicitly account for temporal dependence. However, the DL models could potentially well reduce temporal biases if spatial P data for each hour can be well corrected and downscaled"

*Response: Thank you for your comments. We agree with you. In this study, the DL models considered each hourly P spatial data as a 2D image and did not explicitly account for*

*temporal dependence among images. However, we can argue if the DL models perfectly bias corrected and downscaled each 2D image, the temporal biases can then be reduced. Therefore we have modified this statement in Section 4.4 of the updated manuscript as follows: "The DL models treated each hourly P spatial data as a 2D image and did not explicitly account for temporal dependence between images. We assumed that the DL models could potentially preserve the temporal dependence of observations if the DL models well bias corrected and downscaled each 2D image."*

5. Line 474. Please, elaborate on this. Precipitation events can have different nature and can behave distinctly, depending on atmospheric conditions. How far and under which conditions can be results for the training in one place can used to estimate hourly P in other places where high resolution data are not available

*Response: Thank you for your comments. Using transfer learning to estimate hourly P in other places where high resolution data are not available deserves a separate study since precipitation types are different at different locations. For example, if we train the DL model for bias correcting and downscaling precipitation at a coastal area where convective precipitation is the main type, the trained model may have difficulties to downscale and bias correct precipitation events at locations where dynamic precipitation is the main type. Adding local information into the trained model may improve model performance of transfer learning. There are many questions need to be explored under this topic about transferability under various climate zones and impact of spatial distance, which deserves a separate study. We have included a discussion in Section 5 of the updated manuscript.*

References

Wang, F., Tian, D., Lowe, L., Kalin, L., and Lehrter, J.: Deep learning for daily precipitation and temperature downscaling, Water Resources Research, 57, e2020WR029308, 2021.

Wang, F. and Tian, D.: On deep learning-based bias correction and downscaling of multiple climate models simulations, Climate Dynamics, 1-18, 2022.

Baño-Medina, J., Manzanas, R., Cimadevilla, E., Fernández, J., González-Abad, J., Cofiño, A. S., and Gutiérrez, J. M.: Downscaling multi-model climate projection ensembles with deep learning (DeepESD): contribution to CORDEX EUR-44, Geosci. Model Dev., 15, 6747–6758, https://doi.org/10.5194/gmd-15-6747-2022, 2022.

**Response to RC6**

The article proposes some improvements to the authors model for downscaling precipitation presented in Wang et al, (2021) where they use the loss function MSE instead of MAE. The three proposed improvements are:

- using a weighted MAE as the loss instead of the MAE,

- using a second loss function on a quantized version of the upscaled Stage IV data

- and including other coarse grained predictors.

They evaluate these improvements in the task of downscaling hourly precipitation from the coarse grained MERRA2 ( 50km2) to the fine grained Stage IV ( 4km2) in a rectangle coastal area of the Gulf of Mexico covering the states of Alabama, Mississippi and Louisiana. They evaluate the performance of models by comparing the KGE-score on different aggregations as well as extreme events. The authors conclude that all three of their proposed improvements are helpful. In two marginal notes, the authors evaluate whether coarse-grained predictors make precipitation redundant as an input and whether model performance is related to its complexity. While they state the first to be negative, they state the second to be true.

The problem of downscaling precipitation is relevant and tailoring proven deep learning methods to this problem is a valuable contribution. However, the presented study has severe issues that considerably weaken its interpretation and the possible impact of the study considerably. Further, parts of the manuscripts need major updates. This article requires major revisions before publication.

**General comments**

**Study**

1. Unfortunately, the results presented in Tables 3 and 4 are not enough to estimate the usefulness of the three proposed improvements. The differences between the "Scenarios" 2 to 6 are marginal and the order differs a lot between tasks and metrics. This is especially critical, since the chosen method (a deep neural network) is inherently stochastic and hence, differences between different "Scenarios" might be due to this stochastisity. This stochastisity is further increased by the special training method that the authors use. Instead of presenting each time step once in each epoch, they present random 1897 independent sampled batches of 64 random time steps (which should be mentioned in the manuscript). To distinguish between these random effects and the effect of the proposed improvements it would be necessary to run the models multiple time and assess the significance of the differences between results.

*Response: Thank you for your comments. We agree with you that stochasitisity can play an important role. To address your concern, we have run each scenario three more times (4 in total for each scenario) and evaluated the stochastisity with mean and standard deviation of the four runs. The computational time for each scenario is 20 to 22 hours and running four times for each scenario is feasible for us to consider stochasisity for this study. The results are presented in Table S2 and S3 in the Supplement document, which is dicussed in the discussion section (Section 5) of the updated document. Table S1 indicates that Scenario2 to Scenario6 are significantly different (with p-value 0.05 and confidence*

*interval mean±2\*standard deviation) for hourly precipitation evaluation in terms of KGE metric, which indicates the differences among the six scenarios are not caused by stochastisity. For daily aggregation, Scenario5 and Scenario6 are significantly better than Scenario2 to Scenario4, which is consistent with the results and statement in Section 4.3. For monthly aggregation, Scenario4 are significantly worse than Scenario2, Scenario3, Scenario5 and Scenario6 and no significant differences are found among Scenario2, Scenario3, Scenario5 and Scenario6, which is consistent with the findings in Section 4.3. For extreme indices (see Table S3), Scenario4 is significantly worse than Scenario3, Sceanrio5 and Scenario6 at $99^{th}$ percentile, which is consistent with the findings in Section 4.4. For the annaul maximum wet spell index, Scenario2 and Scenario3 are significantly better than other scenarios.We have modified the statements in the result section (Section 4) to make sure the findings are consistent with the significance evaluation from the total four runs in the updated manuscript. We have included a paragraph in the discussion section on the stochasticity evaluation as follows "Due to the stochastic nature of DL models, we ran each DL scenario for additional three times (four times in total) to evaluate the effects of stochasticity comparing with the added value of each customized component of DL models (see Table S2 and Table S3 in the Supplement). The results show that KGE values for each scenario are significantly different at p-value of 0.05 at hourly time scale, which indicates that the added value of each customized component is not caused by model stochasticity. Scenario1 is significantly worse than other scenarios including QDM_BI at hourly and aggregated time scales as well as extreme indices, emphasizing the added value of weighted loss function. Scenario5 and Scenario 6 are significantly better than other scenarios including QDM_BI in terms of KGE values at hourly and aggregated time scales, and Scenario4 is significantly worse at monthly time scale. For the 99th percentile extreme index, Scenario4 is significantly worse than Scenario3, Sceanrio5 and Scenario6. For the annual maximum wet spell index, Scenario2 and Scenario3 are significantly better than other scenarios. All these stochastic significance evaluation results are consistent with the findings in Section 4. Due to computational demand (20 to 22 hours for running each scenario once) and resource limits, we ran limited times for each scenario to consider stochasticity of DL models and incorporating DL models with Bayesian inference is a potential way to quantify systematic uncertainty caused by model itself as indicated by Vandal et al. (2018a)."*

*Furthermore, we have noted the total number of iterations for each scenario in Section 3.1.5 of the updated manuscript.*

2. Further, to evaluate the three different improvements independently it would be interesting to test and compare all eight possible combinations. The paper, unfortunately, only reports results on five of the eight combinations. Having the three missing combinations (standard MAE + categorical Loss, standard MAE + covariates and standard MAE + categorical Loss + covariates) will help immensely in disentangling the effects of the individual improvements.

*Response: Thank you for your comments. Since we have shown that standard MAE has difficulties to handle highly unbalnaced hourly precipitation data and tends to highly underestimate precipitation (see Table 3, Table 4, and Figures 3 to 8), it is not necessary to test the combination of the standard MAE with other settings.*

3. The only difference that is apparent without the need of a test of significance is the difference between the "Scenarios" that use the weighted MAE and the "Scenario" that use the standard MAE. However, it is unclear, why the authors choose to modify the baseline (Wang et al., 2021) to use a MAE instead of a MSE. In fact, to understand the performance of the proposed model in comparison to the state of the art, a comparison to the original baseline would be very helpful. Especially since the baseline reached a KGE of 0.951 on a slightly different task, which indicates that it might be very competitive. Especially to support claims like "These results highlight the advantages of the customized DL model compared with regular DL models as well as traditional approaches, which provides a promising tool to fundamentally improve precipitation bias correction and downscaling and better estimate P at high resolution."[lines 502 to 505]

*Response: Thank you for your comments. Wang et al. (2021) used regular MSE as loss function, which works well for downscaling daily precipitation through synthetic experiments with no bias, since the precipitation data was first coarsened and then downscaled into the original fine scale. However, in this study we considered two different datasets at hourly scale with large discrepacies (or biases). Particularly, the Stage IV radar observations, as the training target, include outliers (extreme large values). The MSE loss (the square operation) makes the algorithm very sensitive to these outliers (see Ravuri et al., 2021). Ravuri et al. (2021) applied a UNet based architecture (as a baseline) for precipitation nowcasting with radar data and they stated that "We also found that including a mean squared error loss made predictions more sensitive to radar artefacts; as a result, the model is only trained with precipitation weighted mean average error loss.". Here "mean average error" is MAE. Based on the findings from Ravuri et al. (2021), we decided to use MAE intead of MSE as a loss function. We have added the following statement in Section 3.1.4 of the updated manuscript: "Wang et al. (2021) used regular mean squared error (MSE) as loss function, which works well for downscaling daily precipitation through synthetic experiments with no bias, since the precipitation data was first coarsened and then downscaled into the original fine scale. However, in this study, the coarse resolution MERRA2 has substantially biases compared to Stage IV radar data and Stage IV radar data also includes artefacts (e.g., spurious large values) (Nelson et al., 2016). Previous study has shown that MSE loss function is more sensitive to radar artefacts than the mean absolute error (MAE) loss function (Ravuri et al., 2021). Therefore, we chose MAE as a regular loss function in this study."*

4. Since the central result of the paper seems to be that MAE is not a suitable loss for downsampling precipitation, the paper should include a discussion on why someone might consider this to be a sensible idea in the first place, which will amplify the impact of the result. However, at this stage, the motivation for this change in the baseline is unclear from the paper and (to the best of my knowledge) it was not suggested to use MAE in the literature (at least not in the related work presented in the paper).

*Response: Thank you for your comments. We have explained why we chose MAE intead of MSE as a regular loss function in the previous response. Your comment that "MAE is not a suitable loss function for downssampling precipitation" is true for downscaling hour precipitation. Regular MAE may work for downscaling daily precipitation with limited biases (Sha et al., 2020a), but to our knowledge, there are no successful cases using regular MAE for downscaling hourly precipitation due to much higher unbalnace issue (more no*

*rains for hourly than daily). We have added the following explanations in the discussion Section 5 of the updated manuscript: "Regular MAE has been used for downscaling daily precipitation data with limited biases in previous studies (e.g., Sha et al., 2020a), but to our knowledge, there are no successful cases using regular MAE for downscaling hourly precipitation data with large biases."*

5.  Finally, the two side nodes of whether the coarse grained precipitation can be excluded as a predictor ("Scenario4") and whether larger models are overfitting in this specific example do not fit naturally in this study and distract from the main point of the study. Since both of them cannot be answered significantly from the results, I recommend to exclude them to further the readability.

*Response: Thank you for your comments. The reason that we include Scenario4 is to test whether only using covariates are sufficient for estimating hourly P as stated in Section 3.1.5. In addition, the importance of including the bias corrected (achived in MERRA2) coarse grained precipitation is more clear by comparing Scenario4. Furthermore, considering multitask learning is also positive on certain aspects (e.g., the extreme event in Figure 8 and classification result in Figure 9), even though the improvement is not consistent significant due to including more training parameters. With more and more datasets avaiable in the future, including multitask learning concept may work better someday and gives readers more options. In summary, we think including Scenario 4, Scenario3 and Scenario6 can provide useful information for future research.*

**Manuscript**

1.  Many of the above mentioned comments have implications on the manuscript. For example the discussion is quite long and discusses many aspects that are not reflected in the experimental results in any significant way (lines 307-314, 330-340, 349 - 351, 357-362, 367-375, 380-387, 388-392, 405-419, 422-428, 447-479). I recommend to focus the discussion of results mainly on significant results to not "over-interpret" the results and, consequently, "over-claim".

*Response: Thank you for your comments. We have made significant modifications for the manuscript based on the stochastic evaluation to make sure our findings are consistent and do not over interpret the results.*

2.  Further, many of the figures require more work. Figure 1 should maybe reference the very similar figure in (Wang et al., 2021). Many figures have incomplete or no colorbars. Figure 5 and 6 are hard to read and I recommend to exclude them. The interpretation of Figure 9 is unclear.

*Response: Thank you for your comments. We have included the reference of Wang et al. (2021) in the caption of Figure 1. We have added units for the colorbars in the caption of each figure. While it is hard to distinguish 8 lines in one plot, Figures 5 and 6 clearly show the performance of different DL scenarios in comparison with QDM_BI, which provide useful information. We have added more information about the IOU metric in Section 3.3 and included examples to interpret IOU metric in the results section.*

3.  Additionally, it would be helpful if the motivation for each of the three individual contributions is clearly stated in the beginning of the paper.

*Response: Thank you for your comments. We have added following explanations in the introduction section of the updated manuscript: "Traditional DL loss functions have difficulties to handle hourly precipitation data that are highly unbalanced with many zeros and highly positive skewed for nonzero components, therefore, customized DL with weighted loss function to better balance nonzero components has the potential to improve the DL model performance. Besides the primary task of downscaling and bias correction task, adding a highly relevant classification task has the potential to improve DL model performance on the primary task. Incorporating covariates selected based on precipitation formation theory (cloud mass movement and thermodynamics) into DL model also have the potential to improve precipitation downscaling and bias correction. "*

4. Further, the structure of the paper is unclear, for example "Data and methodology" is combined into one section, but is immediately split into two parts, which are data and methodology in 2.1, 2.2. It would be helpful to my understanding of the manuscript to restructure the work.

*Response: Thank you for your comments. We have separated Data and methodology into two sections (Data and Study Area in Section 2, and Methodology in Section 3). The following section numbers are changed accordingly.*

5. The notation of different models as "Scenarios" is confusing.

*Response: Thank you for your comments. We have explicitly described each scenario in Section 3.1.4 of experiment design and in Table 1. We also added scenario settings for Table 3 and Table 4 and made modifications to make them more clear in the revised manuscript.*

6. Often the choice of references is confusing. For example Li et al. (2021) is cited for IoU even though the paper includes no information on IoU that is not also included in this manuscript. This is just an representative example for other cases.

*Response: Thank you for your comments. Li et al. (2021) used intersection over union metric, but they used short name IOU instead of IoU. So we have changed IoU to be IOU in the updated manuscript. We have checked other areas about references inconsistence and made modification as necessary.*

7. Finally, the interpretation of the KGE, more specifically the interpretation of β and γ is surprising. The authors, for example, state that "Scenario1" "highly overestimated the variability"[line 306] however, if we calculate σs/σo = 0.37, indicating, that the variance is actually under estimated.

*Response: Thank you for your comments. The metric $\gamma$ in the modified KGE is defined as a ratio of estimated and observed coefficients of variation (see Eqn.6, $\gamma = \frac{\sigma_s/\mu_s}{\sigma_o/\mu_o}$) instead of $\gamma = \frac{\sigma_s}{\sigma_o}$. Using $\gamma = \frac{\sigma_s/\mu_s}{\sigma_o/\mu_o}$ instead of $\gamma = \frac{\sigma_s}{\sigma_o}$ ensures that the bias and variability ratios are not cross-correlated as stated by Kling et al. (2012). We claimed that Scenario1 highly overestimated the variability (i.e., higher $\gamma$ ) and "variability" means coefficient of variation (i.e., $\sigma_s/\mu_s$) instead of $\sigma_s$. We have made a note in Section 3.3 of the updated manuscript to emphasize the differences and made necessary modifications.*

**Summary**

In summary I believe that the study aims to close a relevant research gap. Further, the proposed method of testing different models with combinations of different improvements is effective. By repeating the experiments to reach significant results, comparing the results to a state-of-the-art baseline and adding more explanation on the motivation of the proposed changes, the paper will be a valuable contribution.

**References**

Kling, H., Fuchs, M., and Paulin, M.: Runoff conditions in the upper Danube basin under an ensemble of climate change scenarios, Journal of Hydrology, 424, 264-277, 2012.

Li, Z., Wen, Y., Schreier, M., Behrangi, A., Hong, Y., and Lambrigtsen, B.: Advancing satellite precipitation retrievals with data driven approaches: Is black box model explainable?, Earth and Space Science, 8, e2020EA001423, 2021.

Ravuri, S., Lenc, K., Willson, M., Kangin, D., Lam, R., Mirowski, P., Fitzsimons, M., Athanassiadou, M., Kashem, S., and Madge, S.: Skilful precipitation nowcasting using deep generative models of radar, Nature, 597, 672-677, 2021.

Sha, Y., Gagne II, D. J., West, G., and Stull, R.: Deep-learning-based gridded downscaling of surface meteorological variables in complex terrain. Part II: Daily precipitation, Journal of Applied Meteorology and Climatology, 59, 2075-2092, 2020a.

Vandal, T., Kodra, E., Dy, J., Ganguly, S., Nemani, R., and Ganguly, A. R.: Quantifying uncertainty in discrete-continuous and skewed data with Bayesian deep learning, Proceedings of the 24th ACM SIGKDD International Conference on Knowledge Discovery & Data Mining, 2377-2386. 2018.

Wang, F., Tian, D., Lowe, L., Kalin, L., and Lehrter, J.: Deep learning for daily precipitation and temperature downscaling, Water Resources Research, 57, e2020WR029308, 2021.

**Response to CEC1 Editor Comments**

Unfortunately, after checking your manuscript, it has come to our attention, it has come to our attention that it does not comply with our "Code and Data Policy".

Your manuscript uses a Deep Learning technique; for it, you use MERRA2 and Stage IV data. You point out the regular repositories for these datasets, but this is too generic information. Learning techniques are hardly reproducible without the specific input and output data files used and obtained. In this way, you must publish the exact input and output data used in your work in one of the repositories we list in our policy. Also, you must reply to this comment with the link and DOI for it and include it in the Data Availability section of your manuscript in any potentially reviewed version you submit.

Please, be aware that failing to comply promptly with this request could result in rejecting your manuscript for publication.

Regards,

Juan A. Añel

Geosci. Model Dev. Exec. Editor

*Response: Thank you for your comments. We have uploaded all the input and output files into the repositories under the three folders named "training", "validation" and "testing", which correspond to the training, validation and testing datasets. The folder "Python code" includes all the python code. The name of each data file corresponding to the name within in the python code (train.py).*